# Spin-valley locking and bulk quantum Hall effect in a noncentrosymmetric Dirac semimetal BaMnSb$_2$

J. Y. Liu [1,8,10], J. Yu [2,9,10], J. L. Ning [1,10], H. M. Yi [2,10], L. Miao [3], L. J. Min [3], Y. F. Zhao [2], W. Ning [2], K. A. Lopez [3], Y. L. Zhu [2], T. Pillsbury [2], Y. B. Zhang [1], Y. Wang [2], J. Hu [4], H. B. Cao [5], B. C. Chakoumakos [5], F. Balakirev [6], F. Weickert [6], M. Jaime [6], Y. Lai [6], Kun Yang [7], J. W. Sun [1], N. Alem [3], V. Gopalan [3], C. Z. Chang [2], N. Samarth [2], C. X. Liu [2✉], R. D. McDonald [6✉] & Z. Q. Mao [1,2,3✉]

Spin-valley locking in monolayer transition metal dichalcogenides has attracted enormous interest, since it offers potential for valleytronic and optoelectronic applications. Such an exotic electronic state has sparsely been seen in bulk materials. Here, we report spin-valley locking in a Dirac semimetal BaMnSb$_2$. This is revealed by comprehensive studies using first principles calculations, tight-binding and effective model analyses, angle-resolved photoemission spectroscopy measurements. Moreover, this material also exhibits a stacked quantum Hall effect (QHE). The spin-valley degeneracy extracted from the QHE is close to 2. This result, together with the Landau level spin splitting, further confirms the spin-valley locking picture. In the extreme quantum limit, we also observed a plateau in the $z$-axis resistance, suggestive of a two-dimensional chiral surface state present in the quantum Hall state. These findings establish BaMnSb$_2$ as a rare platform for exploring coupled spin and valley physics in bulk single crystals and accessing 3D interacting topological states.

[1] Department of Physics and Engineering Physics, Tulane University, New Orleans, LA, USA. [2] Department of Physics, The Pennsylvania State University, University Park, PA, USA. [3] Department of Materials Science and Engineering, The Pennsylvania State University, University Park, PA, USA. [4] Department of Physics, University of Arkansas, Fayetteville, AR, USA. [5] Neutron Scattering Division, Oak Ridge National Laboratory, Oak Ridge, TN, USA. [6] Los Alamos National Laboratory, Los Alamos, NM, USA. [7] Physics Department and National High Magnetic Field Laboratory, Florida State University, Tallahassee, FL, USA. [8] Present address: Department of Physics and Astronomy, University of California, Irvine, CA, USA. [9] Present address: Condensed Matter Theory Center, Department of Physics, University of Maryland, College Park, MD, USA. [10] These authors contributed equally: J. Y. Liu, J. Yu, J. L. Ning, H. M. Yi. ✉email: cxl56@psu.edu; rmcd@lanl.gov; zim1@psu.edu

The combination of inversion symmetry breaking and spin-orbital coupling (SOC) in solid materials provides a route to achieve electronic states with spin polarization in the absence of magnetism. When this occurs in a material possessing valleys in its conduction and valence bands, spin polarization becomes valley-dependent, thus creating a unique electronic state characterized by spin-valley locking. Such an electronic state was first realized in monolayers of group-VI transition metal dichalcogenides (TMDCs) such as $MoS_2$[1–7]. Although the honeycomb lattice structure of bulk 2H-$MoS_2$ is centrosymmetric, it becomes non-centrosymmetric on monolayer. In combination with inversion symmetry breaking, the SOC induced by the heavy element Mo leads to the spin splitting and spin-valley locking of the valence band[1–7]. The spin-valley locked electronic band structure of group-VI TMDC monolayers gives rise to topological valley transport properties such as photo-induced charge Hall effect, valley Hall effect, and spin Hall effect under zero magnetic field[1,8,9], as well as valley-dependent optical selection rule[1–4]. These exotic properties hold a great promise for potential applications in valleytronics, spintronics, and optoelectronics[10].

Although inversion symmetry breaking and SOC can be found in many materials, it is challenging to identify candidate materials which can enable the combination of a valley degree of freedom with inversion symmetry breaking and SOC. Although spin-valley locking has been demonstrated on monolayers of group-VI TMDCs, it is very rarely seen in bulk materials. To the best of our knowledge, there are only two reported examples, i.e., 3R-$MoS_2$[11] and 2H-$NbSe_2$[12]. No materials beyond TMDCs have been reported to show spin-valley locking to date. In this article, we show a previously reported, three-dimensional Dirac semimetal $BaMnSb_2$[13] features unique spin-valley locking. Contrasted with $MoS_2$ monolayer whose spin splitting is large in the valence band (0.15–0.46 eV)[1,11,14,15], but relatively small in the conduction band (1–50 meV)[16], $BaMnSb_2$ shows spin splitting of ~0.35 eV in both conduction and valence bands, much larger than the Dirac gap (~50 meV). In addition, we have also observed a three-dimensional (3D) quantum Hall effect (QHE) in this material, in contrast to the usual two-dimensional (2D) QHE. From the QHE, we have demonstrated the spin-valley locked electronic state has a degeneracy of 2. Moreover, we have also observed a plateau in the z-axis resistance in the quantum limit, which implies a 2D chiral surface metal present in the quantum Hall state.

## Results

**Structure determination.** Previous studies[13,17] suggest $BaMnSb_2$ has a layered tetragonal structure ($I4/mmm$), which is composed of alternative stacking of Sb square net layers and $MnSb_4$ tetrahedral layers, with the Ba layers sandwiched in between the Sb and $MnSb_4$ layers. The Sb layers generate Dirac fermions. Our structural analyses using scanning transmission electron microscopy (STEM) for $BaMnSb_2$ reveal a weak orthorhombic distortion, with the Sb layers forming zig-zag chains, as shown in Fig. 1a, b. Figure 1c shows simulated and experimental high-angle annular dark-field (HAADF)-STEM images along the [100]- and [010]-zone axes. These images indicate the Sb columns within the 2D Sb zig-zag chain layers are evenly spaced along the [010] axis (Fig. 1c upper), whereas the Sb columns' positions shift and form a dimmer-like profile along the [100] axis (Fig. 1c lower). The distortion is highlighted in the magnified experimental HAADF-STEM in Fig. 1d, which is inconsistent with a tetragonal structure where Sb columns should be evenly spaced along both the [100] and [010]-axes, but agrees well with an orthorhombic distortion where Sb atoms on the 2D Sb planes form zig-zag chains, as shown in Fig. 1b. The structure simulation based on first principle DFT calculations finds a non-centrosymmetric orthorhombic structure with the space group of $I2mm$ can well describe the STEM images shown in Fig. 1c. Based on the structural parameters determined by the DFT calculations (Supplementary Table 1), the STEM simulation[18,19] was performed and the simulated STEM images, shown in the left panels of Fig. 1c, match well with the experimental images in the right panels of Fig. 1c. Moreover, the calculated Sb2 shift relative to its position in the tetragonal structure is ~30 pm (Fig. 1b), very close to the Sb2 shift (35 pm) measured in the STEM image (Fig. 1d). The orthorhombic distortion is also evidenced by neutron scattering measurements (see Supplementary Note 1 and Supplementary Fig. 1).

Further, we also conducted optical second-harmonic generation (SHG) polarimetry and microscopy and observed strong SHG signal (Fig. 1e, f), clearly demonstrating the inversion symmetry breaking in $BaMnSb_2$. By rotating the polarization of both the fundamental and the SHG light by 90°, we observed a switch of the contrast in the SHG imaging, indicating the existence of 90° domain walls in the sample. As shown in Fig. 1f, the SHG polarimetry taken in domains A and B can be modeled with $2mm$ point group (twofold axis is along the a-axis in crystal). Details of this model described in Supplementary Note 2 and supplementary Fig. 2 reveal not only the twin walls visible on the surface but also suggest underlying orthorhombic domains below the surface.

**Band structure.** With the $I2mm$ orthorhombic structure, we have calculated the electronic band structure of $BaMnSb_2$. We find the four Dirac nodes along the $\Gamma - M$ line expected for the $I4/mmm$ tetragonal structure phase are completely gaped out for the $I2mm$ structure, while two gapped Dirac cones near the Fermi energy emerge at two different momenta, located symmetrically around the X-point along the X–M line [labeled by valley index $K_{\pm}$ in the inset of Fig. 2a]. The Dirac band dispersion is encircled with the red dashed box in Fig. 2a. The weak interlayer tunneling leads to the two-dimensional nature of the Dirac cones (see Supplementary Note 3). The strong SOC leads to large spin-splitting in each valley, as shown in Fig. 2b where spin projection (i.e., the $\langle S_z \rangle$ value) is color-coded. Here it is worth noting that both conduction and valence bands show similar strength of spin splitting (~0.35 eV). As a result, the valley degrees of freedom is coupled to spin for both conduction and valence bands, in stark contrast with $MoS_2$ monolayer for which the spin splitting is large in the valence band[1,11,14,15], but small in the conduction band as noted above[16].

To verify the calculated band structure, we carried out ARPES measurements on $Ba(Mn_{0.9}Zn_{0.1})Sb_2$ crystals. The reason why we chose this sample for measurements is that it is the least hole-doped. As shown in Fig. 2f, we find linearly dispersed bands cross the Fermi level at two momentum points on the $\bar{M}–\bar{X}$ line and these two crossing points are close to the $\bar{X}$ point. This can also be seen clearly in the constant energy map acquired through the intensity integration over the energy range from −15 to −5 meV, as shown in Fig. 2d where two point-like hole pockets neat $\bar{X}$ are readily discernable. The single crossing point along cut 1 in Fig. 2d, e is consistent with the picture that two Dirac cones at $K_{\pm}$ intersect near $\bar{X}$. In accordance with the calculated band structure (Fig. 2a), linear band crossing points should not appear near $\bar{Y}$. Our observation of crossing points near both $\bar{X}$ and $\bar{Y}$ is due to the existence of 90° domain walls in our samples as mentioned above. Figure 2g plots the calculated band (dotted lines) along $\bar{M}\bar{X}$ together with the measured band, from which we see a good agreement between theory and experiment. Although valley-dependent spin polarization cannot be resolved directly from the current ARPES measurements, the total degeneracy of

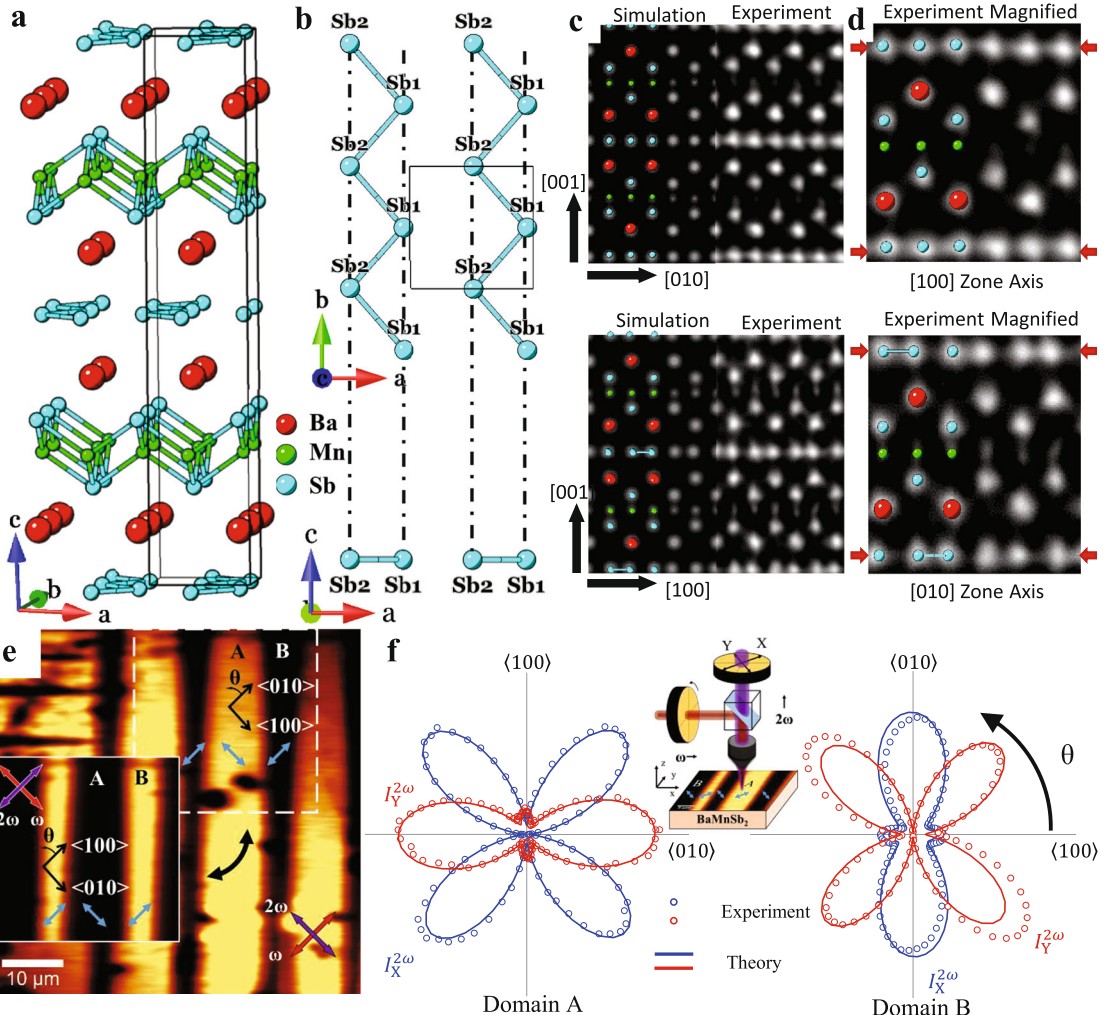

**Fig. 1 Structure determination of BaMnSb₂. a** The schematic showing the *I2mm* crystal structure of BaMnSb₂. **b** Schematic of the Sb zig-zag chain layer from [001] and [010] axis formed due to orthorhombic distortion. **c** Simulated (left panel) and experimental (right panel) HAADF-STEM images along the [100] and [010] zone axes. **d** Magnified experimental HAADF-STEM image with superimposed atom structures highlighting the orthorhombic distortion. The red arrows point to the Sb layer and show how Sb atoms in the zig-zag chain layer form dimmer-like structure in the STEM image taken along the [010]-zone axis. **e** Optical SHG microscopy of polar domains. The polar axes are indicated by the light blue arrows. The imaging light polarization conditions for the ω and the 2ω frequencies are indicated by red and purple arrows. The crystallographic axes in each domain are indicated. The light polarization conditions for the inset image are orthogonal to the corresponding conditions for the main image, hence the reversal in contrast that is observed. **f** SHG polarimetry experiments (circles) in domains A and B indicated in panel (**e**), and the corresponding theoretical fits (solid lines) are shown for two different SHG polarization directions indicated in the inset schematic. Details of the fit based on *2mm* point group symmetry are given in Supplementary Note 1.

~2 extracted from the QHE, which will be discussed below, demonstrates that spin degeneracy in each valley has been lifted, thus leading to the locking of spin and valley degrees of freedom. As for other samples used in this study, including pristine BaMnSb₂ and Eu₀.₁Ba₀.₉MnSb₂, they are more heavily hole-doped. From the quantum oscillation frequencies probed on these samples (see supplementary Table 2), their Fermi energies are estimated to be <0.14 eV, indicating they all have spin-valley locked Dirac cones at $K_\pm$. This is resolved in the ARPES of the Zn-doped sample (Fig. 2g).

To illustrate the physical origin of gapped Dirac cones in BaMnSb₂, we schematically show the band evolution with the orthorhombic distortion and SOC in Fig. 2c. In each Sb layer, there are two Sb atoms within one unit cell, labeled as Sb1 and Sb2 in Fig. 1b. The first principle calculations show that the conduction and valence bands around X (Y) mainly originate from the $p_x, p_y$ orbitals of the Sb2 (Sb1) atom. Therefore, we

construct an 8-band tight-binding (TB) model with 2 sublattices, 2 orbitals ($p_x, p_y$), and 2 spin components, from which we can calculate energy dispersion that captures all qualitative features found in the first-principles calculations (see Supplementary Discussions). To illustrate the band evolution from atomic orbitals with the TB model, we start from the *I4/mmm* structure without SOC, where the conduction and valence bands are close to each other around $\bar{X}$ and $\bar{Y}$ points, which are related by four-fold rotation along $z$. Band crossings between these two bands are found along $\bar{X}$–$\bar{M}$ ($\bar{Y}$–$\bar{M}$), protected by the mirror symmetry $m_x(m_y)$ that only flips $x(y)$ direction (supplementary Fig. 15). The zig-zag distortion that reduces *I4/mmm* to *I2mm* directly gaps out the band crossings near $K_\pm$ by breaking the mirror symmetry $m_x$ (Supplementary Fig. 16), and the resultant two bands at $K$ are from the linear superposition of $p_x$ and $p_y$ orbitals, labeled as $p_\pm$ bands in Fig. 2c. On the other hand, a large gap opens at $\bar{Y}$ for a large distortion (supplementary Fig. 16). Finally, we add the on-

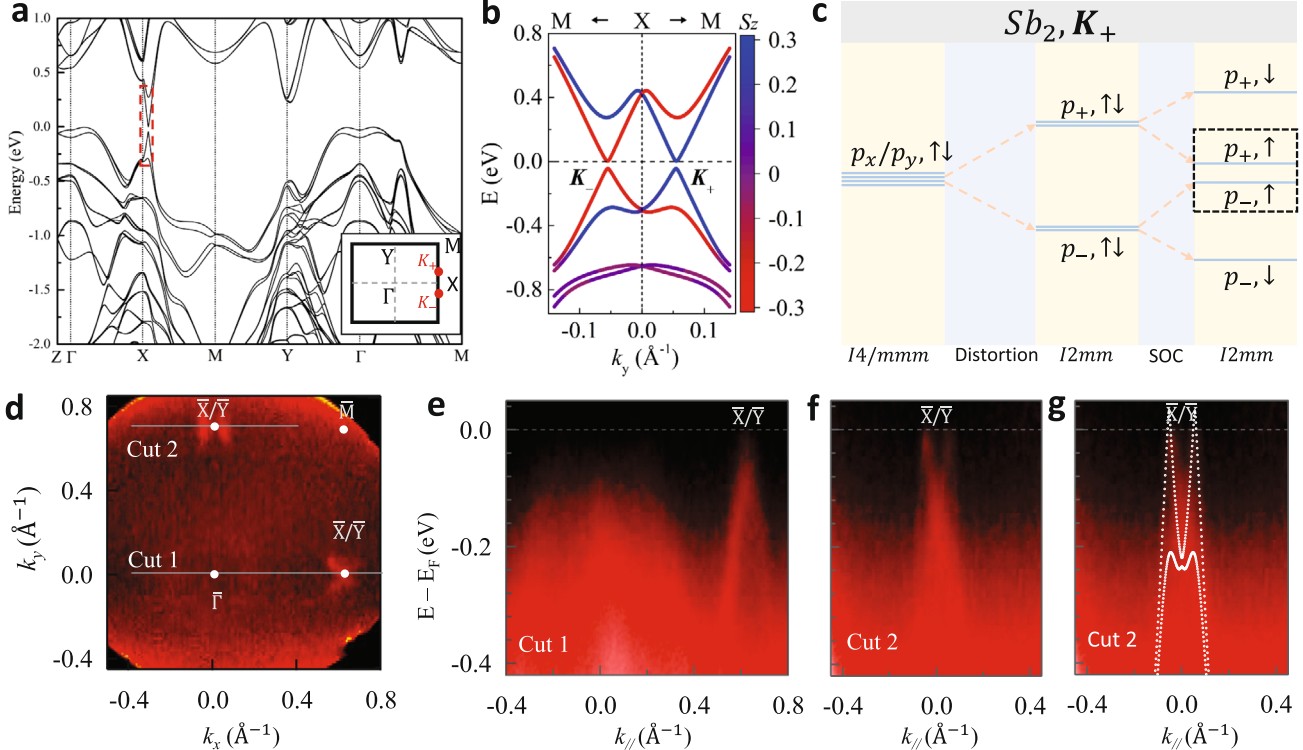

**Fig. 2 Electronic band structure of BaMnSb$_2$. a** The band structure from the first-principles calculation, with the inset showing the first BZ and red dots labeling the two gapped Dirac cones at $K_\pm$. **b** the calculated Dirac band dispersion near $K_+$ and $K_-$, with the spin projection being color-coded (red, spin-up; blue, spin-down). **c** Schematic illustration of the orbital evolution at $K_+$. Here $p_\pm = (p_x \pm ip_y)/\sqrt{2}$; all orbitals come from Sb2 (see Fig. 1b) and the two bases in the black dashed box form the gapped Dirac cone. **d** Constant energy contour of Ba(Mn$_{0.9}$Zn$_{0.1}$)Sb$_2$ on the $k_x$–$k_y$ plane, which is acquired by integrating the intensity from −15 to −5 meV. The ARPES data was taken with the photon energy of 30 eV. Two point-like hole pockets near $\bar{X}$ point can be seen. The presence of hole pockets near $\bar{Y}$ point is due to twin domains. **e** ARPES spectrum along $\bar{\Gamma}\bar{X}$ (cut 1 in panel **d**). There is only one crossing point near the Fermi level ($E_F$) at $\bar{X}/\bar{Y}$. **f** ARPES spectrum along $\bar{M}\bar{X}$ (cut 2 in panel **d**). Two crossing points at $E_F$ can be clearly resolved near $\bar{X}/\bar{Y}$. **g** Comparison of the calculated band (dotted lines) and the band probed by ARPES along $\bar{M}\bar{X}$ (cut 2 in panel **d**).

site SOC, which removes the spin degeneracy of both conduction and valence bands on one valley (Supplementary Fig. 17). Due to strong spin splitting, the bands $|p_+, \uparrow\rangle$ and $|p_-, \uparrow\rangle$ at $K_+$ ($|p_+, \downarrow\rangle$ and $|p_-, \downarrow\rangle$ at $K_-$) are pushed closer to each other and form the gapped Dirac cones with spin-valley locking as mentioned above and illustrated in Supplementary Fig. 18a.

With the understanding of the origin of Dirac cones, we further construct the effective Hamiltonian for the Dirac cone to compare with experiments (see Supplementary Discussions). The effective model is constructed around $K_\pm$ for small $\mathbf{q} = \mathbf{k} - \mathbf{K}$ based on the symmetry and TB model, which reads

$$h_\pm(\mathbf{q}) = \left(E_0 \pm v_0 q_y\right)\tau_0\sigma_0 \pm v_1 q_x\tau_x\sigma_0 \pm v_2 q_y\tau_z\sigma_0$$
$$+ \left[\pm E_1 \pm b_0\left(v_1^2 q_x^2 + v_2^2 q_y^2\right) + v_3 q_y\right]\tau_y\sigma_0 + \lambda_0\tau_y\sigma_z \quad (1)$$

Here $E_0$ and $v_{0,1,2}$ are material related parameters, $\lambda_0$ denotes on-site SOC, $E_1$ and $v_3$ are determined by the distortion, and $\tau$ and $\sigma$ are Pauli matrices for orbital and spin indices, respectively. Without the $\mathbf{q}$-quadratic term, the spin-independent part of the Hamiltonian is in the most general symmetry-allowed form to the first order of $\mathbf{q}$, while the simple symmetry-allowed $\mathbf{q}$-quadratic term is also included to explain the Landau Level splitting as discussed in the next section. From the energy dispersion $E_\alpha =$

$$\left(E_0 + \alpha v_0 q_y\right) \pm \sqrt{\left(v_1 q_x\right)^2 + \left(v_2 q_y\right)^2 + \left(\alpha E_1 + \alpha b_0\left(v_1^2 q_x^2 + v_2^2 q_y^2\right) + v_3 q_y \pm \lambda_0\right)^2}$$ around

$K_\alpha$ with $\alpha = \pm$, the energy gap at $\mathbf{q} = 0$ reads $||E_1| - |\lambda_0||$,

i.e., the difference between SOC ($|\lambda_0|$) and distortion strength ($|E_1|$). By choosing appropriate parameters, we find a good fitting for the energy dispersion between our effective model and the first principles calculation within the experimentally relevant Fermi energy ranges (see Supplementary Fig. 18b). The effective Hamiltonian Eq. (1) explicitly demonstrates the existence of gapped Dirac cones in our system.

**Observation of bulk QHE and Landau level splitting.** The spin-valley locking picture discussed above is further corroborated in our quantum transport measurements. We observed a bulk QHE in BaMnSb$_2$ single crystals due to its layered structure. The inset in Fig. 3e shows an optical image of a 6-electrode Hall bar sample made on a BaMnSb$_2$ single crystal with the thickness of 91 μm (denoted as B#1 below). Figure 3a presents the longitudinal ($\rho_{xx}$) and Hall ($\rho_{xy}$) resistivity data measured on this sample at 1.4 K. Both $\rho_{xx}$ and $\rho_{xy}$ exhibit Shubnikov–de Haas (SdH) oscillations starting from ~3 T and the oscillation frequency $B_F$ obtained from their fast Fourier transform (FFT) analyses is ~18.9 T. When the magnetic field is above 5 T, $\rho_{xy}$ displays clear plateau features while $\rho_{xx}$ reaches minima, implying the presence of bulk QHE in BaMnSb$_2$. The robust evidence for this QHE is given in Fig. 3b which plots $1/\rho_{xy}$ scaled by the step size of the successive $1/\rho_{xy}$ plateaus, i.e., $1/\rho_{xy}^0$ (see the inset to Fig. 3d), as a function of $B_F/B$. When $\rho_{xx}$ reaches minima, $\rho_{xy}^0/\rho_{xy}$ is clearly quantized to half-integer numbers $\gamma = j + 1/2$ ($j$, non-negative integer number), which corresponds to the half-integer normalized filling factor given by $B_F/B$ at the $\rho_{xx}$ minima. All these signatures can be

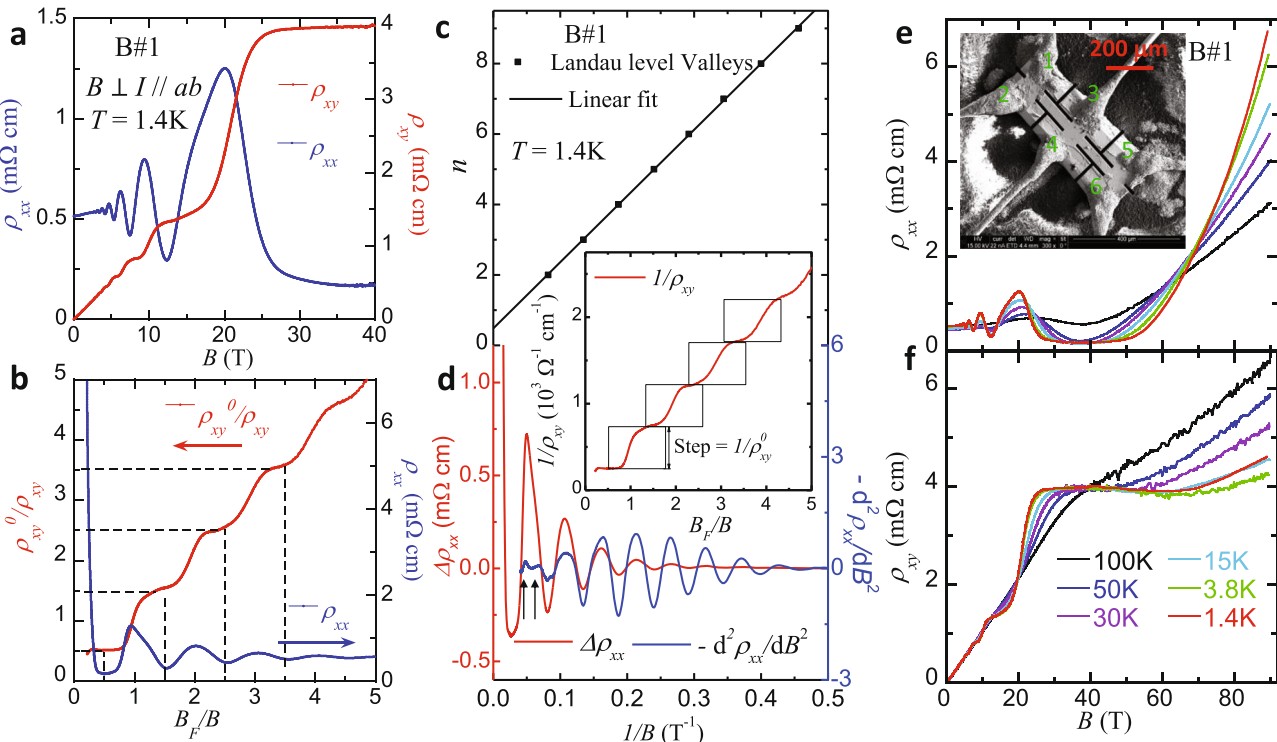

**Fig. 3 Bulk quantum Hall effect in BaMnSb$_2$. a** Magnetic field dependences of in-plane resistivity ($\rho_{xx}$, blue) and Hall resistivity ($\rho_{xy}$, red) at $T = 1.4$ K for the Hall bar sample B#1, measured with field perpendicular to the Sb-plane up to 40 T. Hall plateaus in $\rho_{xy}$ are clearly observed. **b** Normalized inverse Hall resistivity ($\rho_{xy}^0/\rho_{xy}$) and in-plane resistivity ($\rho_{xx}$) versus $B_F/B$. $1/\rho_{xy}^0$ is defined as the step size between the first and the second plateaus of $1/\rho_{xy}$ (see the inset to panel **d**). $B_F$ is the frequency of the SdH oscillations in $\rho_{xx}$. **c** Landau level fan diagram built from the SdH oscillations of $\rho_{xx}$ at 1.4 K. **d** The second derivative, $-d^2\rho_{xx}/dB^2$, which is in phase with $\rho_{xx}$, clearly reveals the oscillation peak splitting near $B = 17$ T. The inset shows the inverse of $\rho_{xy}$ in panel a as a function of $B_F/B$. $1/\rho_{xy}^0$ is determined by the vertical interval between steps. **e**, **f** Field dependences of $\rho_{xx}$ and $\rho_{xy}$ at various temperatures ($T = 1.4$, 3.8, 15, 30, 50, and 100 K), up to 90 T. The inset in (**e**) shows the SEM image of sample B#1 with Hall bar geometry fabricated by FIB.

attributed to a stacked QHE, with the 2D Sb layer sandwiched by the Ba-MnSb$_4$-Ba insulating slabs (Fig. 1a) acting as a quantum Hall layer. The non-trivial Berry phase ($\sim 0.97\pi$) determined from the Landau level (LL) index fan diagram in Fig. 3c confirms the relativistic characteristic of quasi-particles, consistent with the previous reports[13,20]. The QHE observed in sample B#1 persists up to $T = 50$ K, as shown by the temperature dependences of $\rho_{xx}$ and $\rho_{xy}$ in Fig. 3e, f.

We have reproduced such a stacked QHE in several other samples, including Eu- and Zn-doped samples. These samples have various carrier densities and mobilities, as summarized in Supplementary Table 2. Among those samples, the Eu-doped sample E#1 has the highest mobility ($\sim 5040$ cm$^2$/Vs, about three times of that of sample B#1). The QHE of this sample is nearly perfect as compared to other samples, as confirmed by the following observations. First, as shown in Fig. 4a, the $\rho_{xx}$ of this sample corresponding to the $\rho_{xy}$ plateau is very small. For instance, $\rho_{xx}$ is $\sim 0.025$ mΩ.cm for the $\gamma = 3/2\rho_{xy}$ plateau near 20 T (Fig. 4a), one order of magnitude smaller than that of the corresponding quantum Hall state of sample B#1 which occurs near 12.5 T (see Fig. 3a, b). At the $\gamma = 1/2$ quantum Hall state within the quantum limit ($B > 40$ T), its $\rho_{xx}$ becomes much smaller, dropping to zero near 47.5 T, but turning to slightly negative ($\sim -0.015$ mΩ.cm) above 50 T (Fig. 4a). We speculate such a small negative value is induced by the symmetrizing process of the data (see Supplementary Note 5). Second, this sample exhibits not only half-integer quantization in $\rho_{xy}^0/\rho_{xy}$ (Fig. 4c), but also equal steps in the Hall conductivity $\sigma_{yx}$ obtained through tensor conversion from $\rho_{xx}$ and $\rho_{xy}$ (see Supplementary Fig. 3a) and the longitudinal conductivity $\sigma_{xx}$ is nearly zero at the

$\gamma = 1/2$ quantum Hall State (Fig. 4d). In contrast, for samples B#1 and the Zn-doped sample Z#1, although their $\rho_{xy}^0/\rho_{xy}$ shows quantization (see Fig. 3b for B#1 and Supplementary Fig. 4b for Z#1), their $\sigma_{yx}$ does not show equal steps, as shown in Supplementary Fig. 3b and 3c. This is because their $\rho_{xx}$ values at the quantum Hall states are much larger than those of sample E#1. These facts indicate that the QHE is close to a perfect QHE for sample E#1, but imperfect for samples B#1 and Z#1. Imperfect stacked QHE can be ascribed to the inhomogeneous transport due to dead layers (i.e., those Sb layers not showing QHE) and/or imperfect contacts, which is not rare in stacked quantum Hall systems[21]. As will be discussed below, the QHE observed in pristine or Eu/Zn-doped BaMnSb$_2$ originates from the Dirac bands near the X point. Although there exist trivial bands near the Γ point (Fig. 2a and Supplementary Fig. 6), the observation of nearly perfect QHE in sample E#1 suggests that the trivial bands have negligible contribution to transport when the charge carriers hosted by the Dirac bands near the X point have high mobility (see below for further discussions).

Since the bulk QHE in BaMnSb$_2$ is attributed to the parallel transport of 2D Sb layers stacked along the c-axis, the spin valley degeneracy per Sb layer $s$ can be estimated from the step size between the successive $1/\rho_{xy}$ plateaus (i.e., $1/\rho_{xy}^0$, see the inset to Fig. 3d) via the equation $1/\rho_{xy}^0 = sZ^*(e^2/h)$ where $Z^*$ represents the number of layers per unit length[21]. Given each unit cell of BaMnSb$_2$ contains two Sb conducting layers (see Fig. 1a), $Z^* = 1/(c/2)$. Hence $1/\rho_{xy}^0 = s(2/c)(e^2/h)$. The values of $s$ for samples B#1 and E#1 derived from $1/\rho_{xy}^0$ are 1.5 and 2.3, respectively. Measurements on another Eu-doped sample E#3 (Supplementary

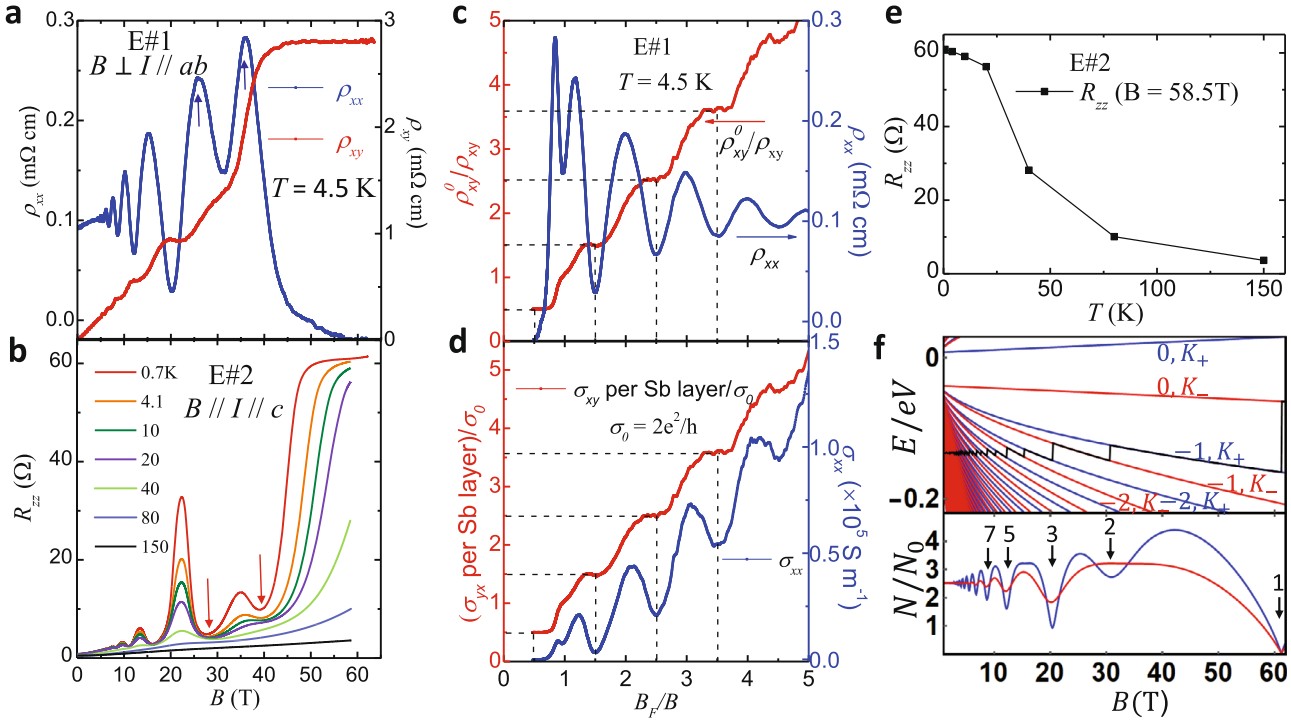

**Fig. 4 Bulk quantum Hall effect in Eu-doped BaMnSb₂. a** Magnetic field dependences of $\rho_{xx}$ and $\rho_{xy}$ of the Hall bar sample E#1, measured with field perpendicular to the Sb-plane up to 64 T (see Supplementary Note 5 and Fig. 11). The arrows indicate the oscillation peak splitting in $\rho_{xx}$. **b** Magnetic field dependence of $R_{zz}$ at various temperatures ($T = 0.7, 4.1, 10, 20, 40, 80,$ and $150$ K), measured on another Eu-doped sample (E#2) with a similar composition. The red arrows indicate the splitting in an oscillation valley. **c** Normalized inverse Hall resistivity ($\rho_{xy}^0/\rho_{xy}$) and in-plane resistivity ($\rho_{xx}$) versus $B_F/B$ for sample E#1. **d** The Hall conductivity $\sigma_{yx}$ per Sb layer normalized by $2e^2/h$, and longitudinal conductivity $\sigma_{xx}$ as a function of $B_F/B$ at 4.5 K. **e** Temperature dependence of $R_{zz}$ at 58.5 T for sample E#2. **f** The LLs calculated from the effective model around $K_\pm$. LLs for $K_+$ and $K_-$ are presented as the blue and red lines, respectively. The black line is $E_F$. Bottom panel in **f**: The DOS at $E_F$ with Gaussian disorder broadening $\Gamma_0 = 2$ meV (blue) and $\Gamma_0 = 3$ meV (red)[23]. The numbers above the black arrows label the LL filling at DOS minima.

Fig. 5) yields $s = 2.2$. For sample Z#1, since its carrier density extracted from the quantum oscillation frequency is about one half of that of sample B#1 (see Supplementary Table 2), its quantum Hall state of $\gamma = 3/2$ can be reached at about 7 T (Supplementary Fig. 4) and the $s$ value estimated for this sample is 1.5. The $s$ values of sample E#1 ($s = 2.3$) and E#3 ($s = 2.2$) agree well with the expected value of $s = 2$ for the spin-valley locking electronic structure shown in Fig. 2b. The small deviation of the experimental values from 2 for these two samples should be due to the errors in the measurements of sample thickness (see "Methods"). For sample E#1 which exhibits a nearly perfect QHE with equal steps in $\sigma_{yx}$ (Supplementary Fig. 3a), we have also estimated its $s$ from the $\sigma_{yx}$ plateau at the quantum limit with $B_F/B = 1/2$ using $\sigma_{xy}/\text{Sb-layer} = (1/2)s(e^2/h)$, where $\sigma_{xy}/\text{Sb-layer}$ is derived from dividing the total Hall conductivity $\sigma_{yx}$ by the sample thickness. The $s$ value obtained through this approach is 2.2, consistent with that estimated from $1/\rho_{xy}^0$ ($s = 2.3$); its deviation from the expected value of 2 is apparently due to the errors in the measurements of sample dimensions. If we take $s = 2$ and normalize $\sigma_{yx}/\text{Sb-layer}$ by $2e^2/h$, the quantized Hall conductivity plateaus can be seen clearly for $\gamma = B_F/B = 1/2, 3/2, 5/2,$ and $7/2$, as shown in Fig. 4d, which further verifies the spin-valley degeneracy of 2 for BaMnSb₂. The smaller $s$ values for samples B#1 ($s = 1.5$) and Z#1 ($s = 1.5$) can be ascribed to the inhomogeneous transport as noted above.

In addition, we have examined the degeneracy via comparing the carrier density estimated from the quantum oscillation frequency $F$ with the transport carrier density extracted from the Hall coefficient. According to Luttinger's theorem, the carrier density of a 2D system with a degeneracy of 2 can be expressed as $n_{2D} =$

$2eF/h$ where $e$ is the elemental charge and $h$ is the Planck's constant. Since one-unit cell in BaMnSb₂ contains two conducting Sb layers (Fig. 1a), the 3D carrier density can be expressed as $n_{SdH} = n_{2D}/(c/2)$, where $c$ is the lattice parameter. As shown in Supplementary Table 2, the estimated $n_{SdH}$ is close to the carrier density determined by Hall coefficients ($n_{Hall}$) for samples E#1, E#3, and B#1. For instance, $n_{Hall}$ for sample E#1 is $1.4 \times 10^{19}$ cm⁻³, and its $n_{SdH}$ is $1.24 \times 10^{19}$ cm⁻³. Such a consistency between $n_{SdH}$ and $n_{Hall}$ gives additional support for the degeneracy of 2. For sample Z#1, we note the difference between $n_{SdH}$ and $n_{Hall}$ is relatively large ($n_{SdH} = 0.36 \times 10^{19}$ cm⁻³ vs. $n_{Hall} = 0.093 \times 10^{19}$ cm⁻³), which can be attributed to inhomogeneous transport caused by Zn doping; that is, those Sb layers exhibiting SdH oscillations and QHE have different carrier density from those layers without showing the SdH oscillations and QHE. $n_{SdH}$ represents only the carrier density of those layers showing SdH oscillations, while $n_{Hall}$ is the average carrier density of all layers.

In the $\gamma = 1/2$ quantum Hall state, we have also observed clear signatures of LL splitting. For sample B#1, this is manifested by the peak splitting in $d^2\rho_{xx}/dB^2$ near $1/B = 0.05$ in Fig. 3d. Such a LL splitting is much more clearly resolved in sample E#1. Although the carrier density of sample E#1 is higher than that of sample B#1 (Supplementary Table 2), we can still observe the quantized $\rho_{xy}$ plateau with $\gamma = 1/2$ in the quantum limit above 40T (Fig. 4c). The $\rho_{xx}$ of this sample exhibits striking splitting (marked by the arrows in Fig. 4a) near $B_F/B = 1$ where $\rho_{xy}$ displays a steep increase (Fig. 4c).

To compare with the QHE observed in experiments, we derive the LLs of Eq. (1) with Peierls substitution $\boldsymbol{q} \rightarrow \boldsymbol{q} + \frac{e}{\hbar}\boldsymbol{A}$ for the external magnetic field $\boldsymbol{B} = \nabla \boldsymbol{A} = (0, 0, B)$ (see Supplementary

Discussion). Given $\lambda_0 E_1 < 0$ from the fitting, the analytical solutions to the leading order are given by $\epsilon_0^{+,\uparrow} = E_0 + E_1 + \lambda_0$ and $\epsilon_0^{-,\downarrow} = E_0 - E_1 - \lambda_0$ for the zeroth LLs. Except for the zeroth LLs, all the other LLs are doubly degenerate due to the valley degeneracy as schematically shown in Supplementary Fig. 7, which accounts for the experimentally observed half-integer filling factor in quantum Hall states (Figs. 3b and 4c). As noted in Fig. 4a, the clear LL splitting has been observed in sample E#1 near 30 T, which may originate from the Zeeman term $\mu_B B \sigma_z$ and/or the $q$-quadratic terms (Supplementary Fig. 7). Since the strong SOC splits the spin degeneracy and locks spins to valleys, the $q$-quadratic terms may also lift the spin-valley degeneracy and cause the LL splitting. Then, we estimate the magnitude of the Zeeman and $q$-quadratic contributions to the LL splitting, and find that the energy scale of $q$-quadratic terms is around 0.3 ($B$/Tesla) meV, much larger than that of Zeeman term $\mu_B B \sim 0.05$($B$/Tesla) meV (assuming the $g$-factor to be 2 here). Thus, this estimate suggests that the $q$-quadratic terms play the major role in inducing the LL splitting. To experimentally evaluate the Zeeman term contribution, we have also measured the angular dependence of the SdH oscillations in interlayer resistance for an Eu-doped sample. As discussed in Supplementary Note 6, the results from these measurements (Supplementary Fig. 12) indeed suggest a weak Zeeman effect and its $g$ factor being much less than that of EuMnBi$_2$ ($g = 9.8(4)$)[22], which provides support for our theoretical assumption of $g = 2$.

The LLs derived from the Hamiltonian with both $q$-quadratic terms and Zeeman coupling are plotted in Fig. 4f and the corresponding density of states is shown in the bottom panel of Fig. 4f, from which we indeed can see the LL splitting for the magnetic field near 30 T, as shown by the blue line in Fig. 4f, in good agreement with the experimental observation shown in Fig. 4a. (Note that we adopted the carrier density of sample E#1 in our calculations in order to compare the LL splitting with this sample.) With increasing the disorder broadening strength[23], the LL splitting disappears and only odd number filling of LLs can be found (see the red curve in Fig. 4f), consistent with the experimental observation of suppressed LL's splitting in a sample (#B1) with lower mobility (Fig. 3a). The consistency between theory and experiment in the LL splitting, together with the $s \sim 2$ value estimated from the $1/\rho_{xy}$ plateau height and $\sigma_{yx}$/Sb layer, provide strong evidence for the spin-valley locking in BaMnSb$_2$.

Although the bulk QHE discussed above bears some similarity with the previously-reported bulk QHE in EuMnBi$_2$[21] where the 2D Bi square-net layers act as QHE layers, the bulk QHE in BaMnSb$_2$ displays distinct features which are absent in EuMnBi$_2$. The QHE in EuMnBi$_2$ is driven by the canted antiferromagnetic (AFM) order produced by the Eu sub-lattice, which reduces interlayer coupling significantly. However, such a canted AFM state exists only in a limited field range (5–22 T), which renders the primary quantum Hall state within the quantum limit inaccessible. In contrast, for BaMnSb$_2$, there is no such a canted AFM order and the presence of its bulk QHE is attributed to intrinsically weak interlayer coupling. With this advantage, we observe the $\rho_{xy}$ and $\sigma_{yx}$ plateau within the quantum limit as discussed above.

**z-axis resistance plateau in the quantum Hall state.** We have also performed measurements on the out-of-plane resistance ($R_{zz}$) as a function of magnetic field at various temperatures using another Eu-doped sample (E#2) with a similar carrier density as sample E#1. The $R_{zz}$ data of sample #E2 are presented in Fig. 4b. We find $R_{zz}$ exhibits a distinct plateau at 0.7 K above 50 T. Such a $R_{zz}$ plateau is a robust feature of the quantum Hall state, as supported by the fact that the $R_{zz}$ plateau occurs concomitantly

with the $\rho_{xy}$ plateau. As shown in Fig. 3f, e, the $\rho_{xy}$ plateau near 40 T as well as the corresponding $\rho_{xx}$ minimum are nearly temperature independent below 20 K, which are typical signatures of a quantum Hall state. The $R_{zz}$ saturation behavior below 20 K (Fig. 4b, e) is consistent with the insensitive temperature dependences of $\rho_{xx}$ and $\rho_{xy}$ below 20 K. This fact, together with the observation of the $z$-axis conductivity $\sigma_{zz}$ and $\sigma_{xx}$ exhibiting the nearly same SdH frequency (supplementary Fig. 10), indicates the $z$-axis transport is dominated by the Dirac bands near the X point and occurs via tunneling process as discussed in Supplementary Note 4.

Despite the observation of trivial hole bands near the $\Gamma$ point using low photon energy ARPES (see Supplementary Fig. 6), the electrical transport is entirely dominated by the X point Dirac bands at all measured magnetic fields and temperatures, indicating that the mobility of the carriers near the $\Gamma$ point are significantly lower than at the X point. The lack of quantum oscillations corresponding to the trivial $\Gamma$ pockets up to magnetic fields approaching 100 T indicates their mobility is at least 2 orders of magnitude lower than the X pockets that exhibit clear Landau quantization at a couple of tesla. Not only do the $\Gamma$ pockets not contribute to the quantum oscillation spectrum, but they are sufficiently localized so as not to have a noticeable impact upon either the low field slope of the Hall effect (that is in good agreement with the single observed SdH frequency from the X-pockets), nor upon the flatness of the quantized Hall plateau. The negligible effect that the $\Gamma$ point band has upon the slope (or their lack of) of the Hall plateau between 40 and 60 T in Fig. 3f and Fig. 4a, further constrains their mobility to be at least 3 orders of magnitude lower than the Dirac bands around the X-Point. As such, we can exclude the possibility that the $R_{zz}$ plateau at the quantum Hall state within the quantum limit is associated with the trivial bands near $\Gamma$. In addition, the possibility of either heating effects or surface accumulation layers due to band bending, which is a phenomenon often seen in narrow gap semiconductors such as InAs[24] and SmB6[25], can also be ruled out, as discussed in Supplementary Note 7.

In general, the role of charge transfer between different Landau quantized bands can have a dramatic effect upon evolution of Landau level population with field and the associated quantum oscillation spectrum. This is particularly apparent in high mobility, low carrier density semimetals such as Bismuth[26] where all bands exhibit robust Landau quantization approaching the quantum limit in high fields. For BaMnSb$_2$, where only one band is orbitally quantized, we can exclude the possibility of charge transfer between the X and $\Gamma$ valleys leading to non-trivial evolution of the Landau level population and quantum oscillation spectrum, as is evident by the periodic quantized Hall conductance in inverse magnetic field in Fig. 4d. Importantly, the above discussions indicate that in the field range where the $R_{zz}$ plateau is observed, the system is in a nearly ideal quantum Hall state at the quantum limit (Fig. 4d), and that the $R_{zz}$ plateau is likely a consequence of the chiral surface state as discussed below.

The $R_{zz}$ plateau exhibits a peculiar temperature dependence: Although the extent of the plateau is reduced in field range with increasing temperature, the extrapolated $R_{zz}$ values near 60 T appears to saturate to a constant for $T < 20$ K. This trend is clearly manifested by the temperature dependence of $R_{zz}$ at 58.5 T (Fig. 4e) and is inconsistent with the generally-expected quantum Hall insulating state, but in line with the 2D chiral surface state expected for a stacked quantum Hall system[27]. Previous studies on semiconductor superlattices have shown that the 2D chiral surface state dominates the $z$-axis transport while the bulk is at the quantum Hall insulating state and this leads the $z$-axis conductivity to be temperature independent below 0.2 K[28]. Given that bulk BaMnSb$_2$ exhibits stacked QHE layers as discussed

above, the $R_{zz}$ plateau as well as its saturation trend below 20 K implies the presence of a 2D chiral surface state on the side wall at the quantum Hall state within the quantum limit. The 2D chiral surface state in the stacked QHE represents a novel quantum liquid comprised of gapless excitations; BaMnSb$_2$ offers an opportunity to explore its underlying physics in bulk single crystals if future experiments further demonstrate its chiral surface state. Although 3D stacked QHE has been observed in several single-crystal materials such as EuMnBi$_2$[21] and ZrTe$_5$[29], the surface chiral metal has not been reported for any of them. Moreover, it is also worth mentioning that the recently-reported bulk QHE in Cd$_3$As$_2$[30] is not a stacked QH system, but arises from the Weyl orbital comprised of the surface Fermi arc on the opposite surfaces of the sample and the 1D chiral LLs in the bulk[31].

## Discussion

The above discussions have shown the Dirac cone at $K_+$ carries only upward spins, while the other one at $K_-$ has downward spins (see Fig. 2b and the inset to Fig. 2a). Although this shares some similarity with the spin-valley locking of the monolayers group-VI TMDCs[1–6], the spin-valley locking in BaMnSb$_2$ exhibits several distinct features. First, the SOC-induced spin splitting at each valley (~0.35 eV) is much larger than the energy gap of Dirac cones (~50 meV) in BaMnSb$_2$, while the TMDC monolayer (e.g., MoS$_2$) is in the opposite limit, with the energy gap (~2 eV) being much larger than the SOC-induced spin splitting (~0.15–0.46 eV for the valence band)[1,11,14–16]. The smaller band gap of Dirac cones compared to the SOC strength in our system implies that the Berry curvature for spin-up state (spin-down state) is more concentrated around the $K_+$ ($K_-$) valley, which can possibly lead to a more strongly coupled valley-spin Hall effect, as compared to TMDC monolayers and bulk 3R-MoS$_2$[11]. Second, the smaller band gap around tens of meV in BaMnSb$_2$ suggests that the optical probe of valley-spin physics will appear in THz frequency regime, rather than the visible light regime. In addition, thanks to the weak interlayer tunneling (see Supplementary Note 4), the valley-spin locking remains in bulk single crystals, contrasted with TMDC materials where the presence of inversion symmetry in bulk material (2H phase) or the films with even number of layers obliterates the valley-spin physics. Finally, the smaller Dirac gap also suggests that the system is close to a topological phase transition and recent theory has suggested that the piezo-electric coefficient varies discontinuously across this transition[32]. Thus, BaMnSb$_2$ will provide a material platform to test this theoretical prediction. Therefore, BaMnSb$_2$ offers a rare opportunity to explore novel spin-valley locking physics, as well as topological phase transition, in bulk materials.

We also emphasize that bulk quantum Hall systems are of particular interest, as exemplified by the observation of novel fractional QHE states in bilayer graphene[33]. Theory has also predicted that if a 3D fractional QHE can be realized in the intermediate tunneling regime of a layered material where interlayer tunneling strength is on the same order of Coulomb energy[34], it can support both new e/3 fermionic quasi-particles capable of freely propagating both along and between layers, as well as new gapless neutral collective excitation modes, i.e., emergent "photon" modes. BaMnSb$_2$ may serve as a playground to test these predictions.

In summary, through structural analyses using STEM, SHG, and first-principle calculations, we find BaMnSb$_2$ possesses a non-centrosymmetric orthorhombic structure with the space group of $I2mm$. From the combined efforts of first principle band structure calculations, tight-binding and effective model analyses, ARPES and transport measurements, we have demonstrated that

the interplay among inversion symmetry breaking, SOC, and valley degree of freedom in BaMnSb$_2$ results in a unique electronic state with spin-valley locking. One distinct feature of the spin-valley locking in bulk BaMnSb$_2$ is that its SOC-induced spin splitting is much greater than the Dirac gap, which may lead to distinct topological valley transport properties. As such, BaMnSb$_2$ provides a rare opportunity to study coupled spin and valley physics in bulk single crystals. In addition, we also observed a plateau in $R_{zz}$ in the quantum Hall state within the quantum limit, which implies the presence of the surface chiral metal state previously predicted for stacked quantum Hall systems.

## Methods

**Single crystals growth for pristine and doped BaMnSb$_2$.** Single crystals of BaMnSb$_2$, Ba(Mn$_{0.9}$Zn$_{0.1}$)Sb$_2$, and (Ba$_{0.9}$Eu$_{0.1}$)MnSb$_2$ used in this study were grown using self-flux method[13]. Prior studies have shown pristine BaMnSb$_2$ is hole-doped[13]; our attempts of doping Zn to Mn sites and Eu to Ba sites aim to tune the chemical potential to reduce hole doping. The Zn-doped sample can indeed reach this goal, but the Eu-doped sample turns out to be even more heavily hole-doped. This is possibly because that the Eu-electron doping is less than the hole doping caused by Mn and Ba vacancies. Supplementary Table 2 shows the comparison of carrier density, mobility, and other parameters among these three types of samples.

**Magnetotransport and sample thickness measurements.** Magnetotransport properties of BaMnSb$_2$, Ba(Mn$_{0.9}$Zn$_{0.1}$)Sb$_2$, and (Ba$_{0.9}$Eu$_{0.1}$)MnSb$_2$ were measured using the National High Magnetic Field Pulsed Field Facility at Los Alamos National Laboratory and the physical property measurement system (Quantum Design). The Hall bar samples of BaMnSb$_2$ and (Ba$_{0.9}$Eu$_{0.1}$)MnSb$_2$ were prepared using focused ion beam (FIB). To achieve homogeneous transport, before we cut the samples using FIB, we first deposited a layer of Au on both edges of the sample and then attached the leads to the Au layers using epoxy.

Since the spin-valley degeneracy $s$ estimated from the QHE is sensitively dependent on the errors in the measurements of sample thickness as discussed in the main text, we carefully measured the sample thicknesses using an optical microscope equipped with a precision ruler. If sample surface is flat enough, the measurement error bar is ~±6 μm. We made best efforts to choose samples with the (001) surfaces being as flat as possible for Hall measurements. From the device images shown in the inset of Fig. 3e and Supplementary Figs. 5a and 11a, no clear terraces can be seen on the surfaces of the samples. Of course, we could not exclude small thickness inhomogeneity. The thickness variation <6 μm is not discernable in the microscope. If we take this into account, the measurement error bar of sample thickness should be between ±6 μm and ±12 μm. As such, for a sample with a thickness of ~100 μm, the sample thickness measurement error should be ≤12%.

**STEM analyses.** Previous work reported BaMnSb$_2$ possesses a centrosymmetric tetragonal structure with the space group of I4/$mmm$[13,17]. As will be shown here, this material indeed involves weak orthorhombic distortion, which is hard to be resolved. We performed comprehensive structural analyses using STEM, neutron scattering, and optical SHG. The STEM sample was prepared using a focused ion beam (FIB) system. Two cross-sectional lamellas perpendicular to each other (oriented in [100] and [010] direction, respectively) were lifted out from the same crystal. The atomic resolution HAADF-STEM images are taken with the Thermo Fisher Titan3 S/TEM equipped with a spherical aberration corrector.

**Optical second-harmonic generation microscopy.** The SHG polarimetry and imaging measurements were performed on a modified Witec Alpha 300S confocal Raman microscope in far-field reflection geometry, using an 800-nm fundamental laser beam generated by a Spectra-Physics SOLSTICE ACE Ti: sapphire femtosecond laser system (pulse width ~100 fs, repetition rate of 80 MHz). A $\lambda/2$ wave plate was utilized to control the polarization direction ($\theta$) of the incident field ($E^\omega$). The second-harmonic field ($E^{2\omega}$) generated through the nonlinear optical process inside the sample was first spectrally filtered, and then decomposed either parallel or perpendicular to the polar axis by an analyzer and finally detected by a photomultiplier tube (PMT). The schematic of our set up is shown in Supplementary Fig. 2.

**Angle-resolved photoemission spectroscopy (ARPES) experiments.** The angle-resolved photoemission spectroscopy (ARPES) measurements were performed at Beamline 5-4 of the Stanford Synchrotron Radiation Light source using a Scienta DA30L electron analyzer. The energy and angle resolutions are ~9 meV and ~0.2°, respectively. The light spot size was set as 36 μm × 26 μm. The samples were cleaved and measured at 15 K in the high vacuum chamber (~5 × 10$^{-11}$ Torr).

**Density functional theory calculations.** The density functional theory[35] calculations are carried out using the Vienna Ab-initio Simulation Package (VASP)[36]. The recently developed strongly-constrained and appropriately-normed (SCAN)

meta-GGA[37,38] is used for its superior performance in description of different chemical bonds and transition metal compounds[37–42]. The projector-augmented wave (PAW) method[43,44] is employed to treat the core ion-electron interaction and the valence configurations are taken as Ba: $5s^26s^25p^6$, Mn: $3p^63d^64s^1$, and Sb: $5s^25p^3$, and an energy cutoff of 520 eV is used to truncate the plane wave basis. We use a $\Gamma$-centered $8 \times 8 \times 1$ mesh K-space sampling for electronic self-consistent calculations within $10^{-6}$ eV per unit cell. Geometries of both the tetragonal phase and the zigzag distorted orthogonal phase of BaMnSb$_2$ were allowed to relax until the maximum ionic forces were below a threshold of 0.001 eV Å$^{-1}$.

## Data availability

All relevant data are available from the corresponding authors upon request. Note: Our original manuscript was posted on arXiv in July 2019 (J.Y. Liu et al., arXiv:1907.06318). Over the course of revising our manuscript, we noted Sakai et al. also reported the study of QHE and spin-valley coupling of BaMnSb$_2$ in a manuscript posted on arXiv in Jan. 2020 (Sakai et al., arXiv:2001.08683)[45].

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

## Acknowledgements

We thank Jainendra Jain for insightful comments. This work was supported by the US Department of Energy under grants DE-SC0019068 and DE-SC0014208 (support for personnel, sample synthesis, high-field measurements, and data analyses); a part of sample synthesis and high-field measurements was supported by the U.S. Department of Energy under EPSCoR grant No. DESC0012432 with additional support from the Louisiana Board of Regents. Work at the National High Magnetic Field Laboratory was supported by National Science Foundation (NSF) DMR-1644779, the State of Florida, and the U.S. Department of Energy (DOE). F.B. and M.J. acknowledge support from the DOE BES 'Science of 100 T' program, R.D.M. and Y.L. acknowledge support from the Center for the Advancement of Topological Semimetals, an Energy Frontier Research Center funded by the U.S. DOE, Office of Basic Energy Sciences. R.D.M. also acknowledges support from the LANL LDRD DR20160085 'Topology and Strong Correlations' for the start of this work. The preliminary ARPES experiment was carried out at the 2DCC-MIP supported by NSF cooperative agreement DMR 1539916. Use of the Stanford Synchrotron Radiation Lightsource, SLAC National Accelerator Laboratory, is supported by the U.S. Department of Energy, Office of Science, Office of Basic Energy Sciences under Contract No. DE-AC02-76SF00515. K.Y.'s work is supported by National Science Foundation grants DMR-1932796 and DMR-1644779. J.H.'s work is supported by the US Department of Energy (support for some of the transport data analyses), Office of Science, Basic Energy Sciences under grant DE-SC0019467. H.B.C. and B.C.C. acknowledge the neutron scattering user facility sponsored by the Scientific User Facilities Division, Office of Basic Energy Sciences, U.S. Departmen of Energy. L.M., K.A.L., V.G., and N.A. are supported by NSF through the Pennsylvania State University Materials Research Science and Engineering Center DMR 2011839. L.M. and N.A. also acknowledge the Air Force Office of Science Research (AFOSR) program FA9550-18-0277 for support. L.M., K.A.L. and N.A's work utilized resources provided by the NSF-MRSEC-sponsored Materials Characterization Lab at Penn State. C.X.L. and J.Y. acknowledge the support of the U.S. Department of Energy (Grant No. DESC0019064) for the development of the

theoretical model, and also the support from the Office of Naval Research (Grant No. N00014-18-1-2793) and Kaufman New Initiative research grant KA2018-98553 of the Pittsburgh Foundation. C.Z.C. acknowledges the support from the NSF-CAREER award (DMR-1847811) and the Gordon and Betty Moore Foundation's EPiQS Initiative (GBMF9063 to C.Z.C.).

## Author contributions

The samples used for this study were synthesized by J.Y.L., Y.L.Z., L.J.M., and Y.W. The Hall bar samples were prepared by R.D.M. and Y.L. using FIB. The transport measurements under pulse magnetic fields were conducted by J.Y.L., R.D.M., F.B., F.W., Y.L., W.N., and M.J. The transport data analyses were made by J.Y.L., J.H., and Z.Q.M. L.X.M., K.A.L., and N.A. did STEM experiments. L.J.M. and V.G. performed the optical SHG measurements. H.M.Y., Y.F.Z., T.P., C.Z.C., and N.S. did ARPES experiments. The neutron scattering experiments were performed by H.B.C & B.C.C. The electronic band structure was calculated by J.L.N., Y.B.Z., and J.W.S. The detailed theoretical analyses were carried out by J.Y. and C.X.L. and K.Y. also contributed to the theoretical interpretation of the bulk QH state. Z.Q.M., J.Y., and C.X.L. drafted the manuscript with contributions from all co-authors. Z.Q.M. and R.D.M. supervised the experimental part of this work and C.X.L. supervised the theoretical part.

## Competing interests

The authors declare no competing interests.
