## [Peer Review File · Nature Communications]

REVIEWER COMMENTS

Reviewer #1 (Remarks to the Author):

The results presented in this paper are very similar to those published in Ref. 42 [Phys. Rev. B 101, 081104(R), (2020)]. The authors' claim that spin-valley locking occurs in BaMnSb₂ is relatively well justified by the filling factor observed in the 3D quantum Hall effect. The arguments presented here are in fact identical to those used in Ref. 42. The material under discussion has a high mobility, and the spin splitting is well resolved. Up to this point, the paper is quite solid.

This reviewer is, however, somewhat skeptical about how the authors come to the conclusion that the saturation of the out-of-plane resistance R_{zz} indicates the presence of a chiral surface state.

Here are some of the questions (and related comments) that come to mind as one considers this manuscript.

1. The strength of the spin splitting seems to depend on the position of the Fermi energy in the theoretical simulations. After the valleys merge at high Fermi energy, one ends up with a single spin-degenerate valley. Since the authors have studied a number of samples with different carrier densities, it would be interesting to know whether any of the samples studied have a sufficiently high Fermi energy to be in this regime?

2. Does the carrier density extracted from quantum oscillations, including the 2-fold degeneracy (1 spin, 2 valleys), agree with the Hall density?

3. The evidence used to demonstrate the presence of chiral surface states is in my view insufficient. Earlier in the manuscript, the authors mention that a band at the gamma point is partially occupied, yielding additional conduction channels. Couldn't that explain the saturation of R_{zz} and the absence of localization at high field? Additionally, given that the material behaves essentially like a narrow-gap semiconductor near the X point, even a slight amount of band bending should lead to surface accumulation layers, giving rise to trivial surface states. These trivial accumulation layers are commonly observed in narrow gap semiconductors, and were recently considered for explaining similar behavior in InAs [ArXiv Jaoui et al., 2008.06356] and SmB₆ [Hlawenka et al., Nat. Comm. 9, 517 (2018)].

Additionally, the authors should define ρ_{xy}^0 as soon as it comes up in the text. And in the author list, should the name Mcdonald be spelled McDonald?

The authors should address the points raised above (and particularly item 3) before the paper can be considered for publication in Nature Communications.

Reviewer #2 (Remarks to the Author):

This work reports a comprehensive study of BaMnSb₂. Based on theoretical calculations, angle-resolved photoemission spectroscopy and quantum transport measurements, the authors claim that the energy bands around the Fermi level are dominated by a pair of symmetric spin-split bands, following a gapped Dirac dispersion. The splitting is of similar strength in both the valence band and

the conduction band. I find that the arguments are convincing and the material is interesting in some aspects in comparison with thin CMD films. I recommend publication of the manuscript.

I have a few comments to be addressed by the authors.

In this work, the key evidence of spin-valley locking, although probably not a direct one, is the degeneracy of the Dirac band being 2 instead of 4. The degeneracy is estimated from the value of the last Hall plateau. Unlike the quantum Hall effect in 2D, where the quantized resistance does not depend on the geometry, such an estimation in 3D is prone to error. The problem is particularly severe when the longitudinal resistance does not go to zero, as one has a freedom to choose between the conductance and resistance. So, it often raises concerns. Therefore, it is necessary to be thorough. For instance, can equal steps be observed if the Hall conductivity is presented in Fig. 3b? What would be the degeneracy if the conductivity is used for Z#1, which has a large longitudinal resistivity? Why does the longitudinal resistivity become negative at high fields, as seen in Fig. 4a? Will it have any effect on obtaining the correct Hall resistivity?

I would like to point out that the degeneracy can be obtained by comparing the Hall resistivity, which gives the carrier density, with Bf , which tells the size of the Fermi surface. This is not entirely an independent method, still it is good to show, as it does not have the dilemma of which to choose between resistivity or conductivity.

As for the plateau in σ_{zz} (Fig. 4b), is it possible that this is due to joule heating when the resistance is strongly enhanced?

We thank both referees for reviewing our manuscript. We appreciate their comments and suggestions, which have been very helpful in improving our manuscript. In the following, we provide a point-to-point response (shown in blue) to all comments raised by the two referees (shown in black).

Response to Reviewer #1

The results presented in this paper are very similar to those published in Ref. 42 [Phys. Rev. B 101, 081104(R), (2020)]. The authors' claim that spin-valley locking occurs in BaMnSb₂ is relatively well justified by the filling factor observed in the 3D quantum Hall effect. The arguments presented here are in fact identical to those used in Ref. 42. The material under discussion has a high mobility, and the spin splitting is well resolved. Up to this point, the paper is quite solid.

Response: We appreciate the referee's judgment: "the paper is quite solid". While we acknowledge the main argument made in our manuscript seems similar to that of Ref. 42, we must point out our work has several aspects distinct from Ref. 42, which makes our arguments solid, as discussed below.

First, the determination of the non-centrosymmetric orthorhombic $I2mm$ structure plays a critical role in establishing the spin-valley locked Dirac state of BaMnSb₂. This material was previously reported to be tetragonal with the space group of $I4/mmm$ (Z. Naturforsch. 32b, 383 (1977)). The neutron scattering data obtained in our previous work (Scientific Reports 6, 30525 (2016)) can also be refined using the reported tetragonal structure. These facts imply that either X-ray or neutron diffraction is hard to resolve the $I2mm$ orthorhombic structure of BaMnSb₂, which is indeed reflected in our significant efforts made to the structure determination of BaMnSb₂.

Although we observed bulk quantum Hall effect (QHE) of BaMnSb₂ in 2016, we did not rush to publish this data, since we encountered a challenging problem: the spin-valley degeneracy extracted from the QHE is 2, which contradicts the reported $I4/mmm$ tetragonal structure. This is because the tetragonal structure should come with four Dirac nodes, with the spin-valley degeneracy of 8 according to a previous prediction (J. Phys. Condens. Matter 26, 042201 (2014)). This inconsistency motivated us to carefully reexamine the crystal structure. Since the orthorhombic distortion of BaMnSb₂ is very small, it is extremely difficult to determine its orthorhombic space group using either neutron scattering or single crystal X-ray diffraction. Two of the co-authors of our manuscript, Dr. Huibo Cao and Dr. Bryan Chakoumakos at the Oak Ridge National lab, are the leading experts in determining crystal structures using neutron scattering and single crystal X-ray diffractions. They did both neutron diffraction and single crystal X-ray diffraction measurements on BaMnSb₂ and found the diffraction data can be refined using either the reported $I4/mmm$ tetragonal structure (Scientific Reports 6, 30525 (2016)) without considering a few observed weak forbidden peaks in x-ray diffraction or orthorhombic structures with various space groups (including $I2mm$, $Pmmm$, $Cmmm$) with those few peaks considered. Since they could not finalize the structure from the diffraction data, they suggested us to use other techniques to examine the structure of this material. As shown in the manuscript, we used scanning transmission electron microscopy (STEM) and optical second

harmonic generation (SHG) measurements to finalize the $I2mm$ structure of BaMnSb_2 . Our STEM analyses reveal direct evidence of orthorhombic distortion (Fig. 1c and 1d in the manuscript). However, we were still unable to determine the space group based on the STEM images alone. Since there are several possible orthorhombic space groups as noted above, we performed first principles calculations, from which we found the $I2mm$ orthorhombic structure is the most stable. The $I2mm$ structure is non-centrosymmetric. If this is correct, it should generate SHG response. To verify that, we conducted SHG measurements and indeed observed strong SHG signal (Fig. 1e and 1f); further we also found the SHG polarimetry can be modeled with the $2mm$ point group (Fig. 1f). These results provide solid evidence for the non-centrosymmetric $I2mm$ orthorhombic structure of BaMnSb_2 . **The spin-valley locked band structure predicted by first principles calculations is built on this determined structure.** Given the critical role played by our structure determination, we have included one section “Structure determination” in the main text to show how the $I2mm$ structure of BaMnSb_2 is determined and why this determined structure is different from the previously reported tetragonal structure.

In contrast, the $I2mm$ structure reported in Ref. 42 was determined only based on single-crystal X-ray diffraction refinement. Given the neutron and X-ray diffraction data can also be refined with other orthorhombic space groups such as $Pmmm$ and $Cmmm$ as noted above, it is clearly risky to claim the $I2mm$ structure for BaMnSb_2 based on the X-ray diffraction structure refinement alone. Without solid evidence for the $I2mm$ non-centrosymmetric orthorhombic structure, their claim of spin-valley locking based on first principles calculations is clearly skeptical. We note that our original manuscript ([arXiv:1907.06318](https://arxiv.org/abs/1907.06318)) was posted in arXiv six months earlier than ref. 42 ([arXiv:2001.08683](https://arxiv.org/abs/2001.08683)) as mentioned in the Note at the end of manuscript, which means our structure determination work was fully independent.

Second, to reveal experimental evidence for the spin-valley locked Dirac state, it is critical to show there are two gapped Dirac cones symmetrically located near Y point along the YM line (see the calculated Fermi surface in Fig. 1e in Ref.42). However, this is not resolved in the ARPES measurements reported in ref. 42. To observe these two Dirac crossing points, the band dispersion along the MY line must be shown (see Fig. 1e in Ref. 42). However, Ref.42 did not show this data. Instead, they presented the band dispersion data only along the Γ -Y line, which cannot reveal two Dirac cones along the MY line. Although their constant energy map (Fig. 1d in Ref. 42) appears to show two spots near Y, it is hard to claim they correspond to two Dirac band crossing points without showing the band dispersion along the YM line.

In sharp contrast, in our ARPES measurements, we clearly observed two Dirac crossing points symmetrically residing near X/Y point along the M-X/Y line (Fig.2f in our manuscript). Our constant energy map shown in Fig. 2d clearly demonstrates two point-like hole pockets comprised of linear bands near X/Y point, which is a clear evidence of a paired Dirac cones near X/Y point (Note that X/Y points cannot be distinguished due to the coexistence of twin domains in the orthorhombic structure). Then, through the bulk quantum Hall effect (QHE) observed in BaMnSb_2 , we demonstrate the total spin-valley degeneracy of this Dirac state per Sb layer is two, indicating the spin degeneracy is lifted for both Dirac cones, thus resulting in spin valley locking as shown in Fig. 2b. Without clear ARPES evidence for the paired Dirac cones, it would be difficult to claim the spin-valley coupled Dirac state.

Third, the claim of QHE in Ref. 42 is not well justified for two reasons: (a) the normalized inverse Hall resistivity ρ_{xy}^0/ρ_{xy} (see Fig. 3a in Ref. 42) is not equally spaced and ρ_{xy}^0/ρ_{xy} deviates from half integer numbers 5/2 and 7/2 for $B_F/B = 5/2$ and 7/2. In contrast, in our work, ρ_{xy}^0/ρ_{xy} is quantized to 1/2, 3/2, 5/2 and 7/2, respectively, for $B_F/B = 1/2, 3/2, 5/2$ and 7/2 (Fig. 3b in our manuscript). (b) Ref. 42 did not demonstrate the longitudinal resistivity ρ_{xx} reaches a minimum or is close to zero at quantum Hall states, since they were not able to measure ρ_{xx} due to its mixing with the z -axis resistivity ρ_{zz} . In contrast, in our work, we succeeded in measuring ρ_{xx} without a ρ_{zz} component using samples prepared through focused ion beam (FIB) cutting. We indeed find ρ_{xx} reaches a minimum at the quantum Hall states and is very small for the high mobility sample E#1 (see our response to referee 2, page 9-11). Moreover, we also find the SdH oscillations in ρ_{xx} and ρ_{zz} are out-of-phase, indicating the z -axis transport is dominated by tunneling (see section I.4 in supplementary materials). This is not mentioned in Ref. 42.

Finally, we would like to emphasize that we also theoretically corroborated the spin-valley locked Dirac state in BaMnSb₂. Our effective Hamiltonian analyses explicitly demonstrate the existence of spin-valley locked Dirac cones near X. To compare with the QHE observed in experiments, we have also calculated Landau level (LL) spectra and successfully predicted the LL splitting due to the lifted spin-valley degeneracy. These theoretical investigations, which are not included in Ref. 42, deepen our understanding of the physics of the spin-valley locked Dirac state.

This reviewer is, however, somewhat skeptical about how the authors come to the conclusion that the saturation of the out-of-plane resistance R_{zz} indicates the presence of a chiral surface state.

Response: For a stacked quantum Hall system, previous theoretical studies by Balents and Fisher (PRL 76, 2782 (96)) predicted that if the interlayer tunneling amplitude is small compared to the quantum Hall gap, coupling the edge states by interlayer tunneling leads to a 2D gapless chiral surface state on the sidewalls as illustrated in Fig. R1(a) attached below, which is 2D analog of the 1D chiral edge state of a single quantum Hall layer. Such a 2D chiral surface state dominates the z -axis transport within the quantum Hall states, resulting in plateaus in the z -axis conductivity (σ_{zz}) or resistivity (ρ_{zz}). In contrast, between quantum Hall transitions, the z -axis transport is through bulk due to the bulk extended states, resulting in peaks between the σ_{zz} plateaus (see the dashed curve in Fig. R1(b)).

According to this theory, the σ_{zz} plateaus due to the 2D chiral surface state are unquantized, but much less than e^2/h and independent of temperature. These predictions were first demonstrated in the stacked quantum Hall states of the semiconductor superlattices, as shown in Fig. R1(c). In our experiment, we did observe a plateau in the z -axis resistance R_{zz} in the field range where the Hall resistivity plateau with the filling factor of $\nu = 1/2$ is present at 0.7K (see the red curve in Fig. 4b in the manuscript). Further, we find the R_{zz} value near 60T tends to saturate when the temperature is lowered below 20K (Fig. 4b and 4e). Given we have demonstrated BaMnSb₂ is a stacked quantum Hall system and its z -axis transport is dominated by the bulk Dirac bands via a tunneling process (see section I.4 in SM), the observation of the R_{zz} plateau within the quantum Hall state near 60T, together with the saturation behavior of R_{zz} below 20K, are consistent with

the expectation of a 2D chiral surface state in stacked quantum Hall systems. Nevertheless, we agree with the referee that further experiments are needed to verify the chiral surface state in BaMnSb₂, which is the goal of our future work. What we want to emphasize here is that our current z-axis transport data give a strong hint for the presence of chiral surface states. We have revised the manuscript to reflect that and remove “chiral surface state” from the title. Since the major finding of this manuscript is the spin-valley locked Dirac state and quantum Hall state in bulk single crystals, this revision does not change the major significance of this work.

[Redacted]

Figure R1. (a) Schematic of the 2D chiral surface state for a stacked quantum Hall sample. The chiral surface state dominates the z-axis transport, which occurs via tunneling; t represents tunneling magnitude. $V(x,z)$ represents the random potential. (b) Predicted behavior of the z-axis conductivity in stacked quantum Hall systems for an isolated transition (solid line) and with an intervening metallic phase (dashed curve). Panel (a) and (b) are adopted from the paper by Balents and Fisher (PRL 76, 2782 (96)). (c) The magnetic field dependence of in-plane longitudinal and Hall resistance (upper panel) and z-axis conductivity of a stacked quantum Hall system composed of semiconductor superlattice, adopted from the paper by Druist et al., (PRL 80, 365(1998)).

Here are some of the questions (and related comments) that come to mind as one considers this manuscript.

1. The strength of the spin splitting seems to depend on the position of the Fermi energy in the theoretical simulations. After the valleys merge at high Fermi energy, one ends up with a single spin-degenerate valley. Since the authors have studied a number of samples with different carrier densities, it would be interesting to know whether any of the samples studied have a sufficiently high Fermi energy to be in this regime?

Response: Although the samples used in this study indeed show a large variation in carrier density, as summarized in Table S1 in Supplementary Materials, none of them has high enough Fermi energy for the valleys to merge. Among the samples listed in Table S1, sample E#2 has the largest quantum oscillation frequency $F(=34.4\text{T})$, indicating this sample should have the highest Fermi energy. Given that F is directly linked to the extremal Fermi surface cross-section area A_F by the Onsager relation $F = (\Phi_0/2\pi^2)A_F$, we can estimate the Fermi wavevector k_F from F

using $A_F = \pi k_F^2$. The estimated k_F is 0.039 \AA^{-1} for sample E#2. Fig. R2 attached below shows the ARPES spectra probed on the 10%-Zn doped sample which has the lowest Fermi energy. According to the estimated k_F , we can see the Fermi energy of sample E#2 is $\sim 0.14 \text{ eV}$ below that of sample Z#1 and the valleys do not merge at this energy. We have added this information to the revised manuscript (page 7).

Figure. R2: ARPES spectrum along $\overline{M\bar{X}}$ for the 10% Zn doped sample (i.e. Fig. 2g in the manuscript). The Fermi energy of sample E#2, $E_{F,E\#2}$, is denoted by the dashed green line in this figure.

2. Does the carrier density extracted from quantum oscillations, including the 2-fold degeneracy (1 spin, 2 valleys), agree with the Hall density?

Response: We have estimated the carrier density from the quantum oscillation frequency F and compared it with transport carrier density extracted from the Hall coefficient. According to Luttinger's theorem, the carrier density of a 2D system with a degeneracy of 2 can be expressed as $n_{2D} = 2eF/h$ where e is the elemental charge and h is the Planck's constant. Since one-unit cell in BaMnSb_2 contains two conducting Sb layers (Fig. 1a), the 3D carrier density can be expressed as $n_{SdH} = n_{2D}/(c/2)$, where c is the lattice parameter. As shown in Table S1, the estimated n_{SdH} is close to the carrier density extracted from Hall coefficients (n_{Hall}) for samples E#1, E#3 and B#1. For instance, n_{Hall} for sample E#1 is $1.4 \times 10^{19} \text{ cm}^{-3}$, while its n_{SdH} is $1.24 \times 10^{19} \text{ cm}^{-3}$. Such a consistency between n_{SdH} and n_{Hall} gives additional support for the degeneracy of 2. For sample Z#1, we note the difference between n_{SdH} and n_{Hall} is relatively large ($n_{SdH} = 0.36 \times 10^{19} \text{ cm}^{-3}$ vs. $n_{Hall} = 0.093 \times 10^{19} \text{ cm}^{-3}$), which can be attributed to inhomogeneous transport caused by Zn-doping; that is, those Sb layers exhibiting SdH oscillations and QHE have different carrier density from those layers without showing the SdH oscillations and QHE. n_{SdH} represents only the carrier density of those layers showing SdH oscillations, while n_{Hall} is the average carrier density of all layers. We have added these discussions to the revised manuscript on page 11-12.

3. The evidence used to demonstrate the presence of chiral surface states is in my view insufficient. Earlier in the manuscript, the authors mention that a band at the gamma point is partially occupied, yielding additional conduction channels. Couldn't that explain the saturation of R_{zz} and the absence of localization at high field?

Response: We have partially addressed this issue on pages 3-4. We agree with the referee that more experiments are needed to verify the chiral surface state. The R_{zz} plateau in the quantum Hall state as well as its saturation behavior below 20K observed in our current experiment are only suggestive of the 2D chiral surface state of a stacked quantum Hall system. Nevertheless, although the band at the Γ point has small contribution to transport, the saturation of R_{zz} cannot be explained by the band near the Γ point for the following reasons:

First, we measured Hall resistivity ρ_{xy} as a function of magnetic field at various temperatures for sample B#1 and found the background of the field dependence of ρ_{xy} nearly shows temperature independent linear behavior if we do not consider the ρ_{xy} plateau above 20T (see Fig. 3f in the manuscript), indicating the in-plane transport is dominated by a single band despite the existence of the band near the Γ point. Second, if the band near the Γ point made observable contribution to the z -axis transport, ρ_{xy}^0/ρ_{xy} (Fig 4c) and Hall conductivity σ_{yx} (Fig. 4d) would not show quantization, since its contribution to the Hall effect has no reason to be quantized. In other words, the quantization of ρ_{xy}^0/ρ_{xy} (Fig 4c) and σ_{yx} (Fig. 4d) seen in sample E#1 is strongly suggestive of the negligible contribution of the band near the Γ point to the z -axis transport. Third, the carrier density extracted from the Hall efficient for sample E#1 is $1.4 \times 10^{19} \text{ cm}^{-3}$, comparable to the carrier density estimated from the quantum oscillation frequency $1.24 \times 10^{19} \text{ cm}^{-3}$. Given we have demonstrated that both the SdH oscillations and QHE observed in our experiments originate from the linear Dirac band near the X point, the consistency of carrier densities extracted from the SdH oscillations and Hall coefficient indicates that it is the Dirac band near the X point that dominates the transport properties. Additionally, we note the SdH oscillations of ρ_{xx} and R_{zz} (Fig. 4a and 4b) have the nearly same frequency, but out-of-phase, indicating the transport along the z -axis is through a tunneling process. This is because that in the high field region, the ρ_{xx} valleys occur at the quantum Hall states where the density of state reaches a minimum; if tunneling dominates the z -axis transport, the interlayer tunneling conductance should reach a minimum at the quantum Hall state, i.e. a maximum in R_{zz} . This is exactly what we have observed. The detailed discussions on the z -axis tunneling are presented in section I.4 in SM.

From the temperature dependences of ρ_{xx} and ρ_{xy} for sample B#1 (Fig. 3e and 3f), we can see the ρ_{xy} plateau near 40T (Fig. 3f) as well as the corresponding ρ_{xx} minimum (Fig. 3e) are nearly temperature independent below 20K, which are typical signatures of a quantum Hall state. The R_{zz} saturation behavior below 20K occurs concomitantly with the insensitive temperature dependences of ρ_{xx} and ρ_{xy} . This further indicates the z -axis transport is dominated by the Dirac band near the X point and occurs via tunneling process. Therefore, it is less likely that the R_{zz} saturation behavior below 20K (Fig. 4c) is caused by the band near the Γ point. We have added some of the above discussions to the revised manuscript on page 15.

Additionally, given that the material behaves essentially like a narrow-gap semiconductor near the X point, even a slight amount of band bending should lead to surface accumulation layers, giving rise to trivial surface states. These trivial accumulation layers are commonly observed in narrow gap semiconductors, and were recently considered for explaining similar behavior in InAs [ArXiv Jaoui et al., 2008.06356] and SmB6 [Hlawenka et al., Nat. Comm. 9, 517 (2018)].

Response: we thank the referee for bringing these two papers to our attention. Our discussions presented above have shown the z -axis transport is dominated by the bulk Dirac band near the X point and occurs via a tunneling process. If trivial accumulation layers due to band bending existed in BaMnSb₂, they would be present at the top and bottom surfaces along the z -direction, thus not contributing to the z -axis transport. On the other hand, if we assume the z -axis transport was associated with the trivial surface state, the SdH oscillations seen in R_{zz} would not be coupled to the bulk quantum Hall state, which clearly contradicts our experimental observation of R_{zz} reaching maxima at the ρ_{xy} plateaus. As mentioned above, the SdH oscillations of R_{zz} and ρ_{xx} have the nearly same oscillation frequency for the samples taken from the same batch and the carrier densities extracted from the Hall coefficient and quantum oscillation frequency are consistent. These facts further indicate the trivial surface states due to band bending are not involved in BaMnSb₂. In the revised manuscript, we have added more discussions to page 15 in the main text and Section I. 7 in SM to exclude the possibility of the band-bending induced surface accumulation layers.

Additionally, the authors should define ρ_{xy}^0 as soon as it comes up in the text. And in the author list, should the name Mcdonald be spelled McDonald?

Response: To make the definition of ρ_{xy}^0 clear, we have moved Fig. S3 to Fig. 3d as an inset, from which the definition of $1/\rho_{xy}^0$ can be seen clearly. We also corrected the typo in the author list.

The authors should address the points raised above (and particularly item 3) before the paper can be considered for publication in Nature Communications.

Response: We hope we have satisfactorily addressed all the issues raised by the referee. Again, we thank the referee for insightful comments, which have been very helpful in improving our manuscript.

Response to Reviewer #2

This work reports a comprehensive study of BaMnSb₂. Based on theoretical calculations, angle-resolved photoemission spectroscopy and quantum transport measurements, the authors claim that the energy bands around the Fermi level are dominated by a pair of symmetric spin-split bands, following a gapped Dirac dispersion. The splitting is of similar strength in both the valence band and the conduction band. I find that the arguments are convincing and the material is interesting in some aspects in comparison with thin CMD films. I recommend publication of the manuscript.

Response: We thank the referee for the concise summary of our work and the positive assessment of our work. We also appreciate the referee's questions, which have been extremely helpful in improving the manuscript.

I have a few comments to be addressed by the authors.

In this work, the key evidence of spin-valley locking, although probably not a direct one, is the degeneracy of the Dirac band being 2 instead of 4. The degeneracy is estimated from the value of the last Hall plateau. Unlike the quantum Hall effect in 2D, where the quantized resistance does not depend on the geometry, such an estimation in 3D is prone to error. The problem is particularly severe when the longitudinal resistance does not go to zero, as one has a freedom to choose between the conductance and resistance. So, it often raises concerns. Therefore, it is necessary to be thorough.

Response: We agree with the referee that the spin-valley degeneracy s estimated from the 3D stacked quantum Hall effect (QHE) has error, since the estimated s depends on not only the errors in sample dimension measurements, but also the homogeneity of transport. If a certain number of Sb layers do not show QHE (i.e. dead layer) due to disorders/defects and/or imperfect contact, QHE would be imperfect and s would be underestimated. Because of this, we performed QHE measurements on multiple samples with various mobilities and found the errors could lead s to vary in the 1.5-2.3 range, which points to the intrinsic degeneracy of 2 rather than 4. Among the samples we measured, we also found a high-mobility sample (E#1) which exhibits a nearly perfect QHE with both ρ_{xx} and σ_{xx} being close to zero in the quantum Hall state within the quantum limit. The s value of this sample is indeed close to 2, as discussed below.

As shown in the manuscript, we have estimated s using $1/\rho_{xy}^0 = sZ^*(e^2/h)$, where $Z^* = 1/(c/2)$ represents the number of quantum Hall layers per unit length and $1/\rho_{xy}^0$ is the step size between the successive $1/\rho_{xy}$ plateaus (see the inset to Fig. 3d). Given the lattice parameter c can be precisely measured, the error of the estimated s is mainly determined by the measurement error of the sample thickness. We measured the thicknesses of samples using an optical microscope equipped with a precision ruler. If sample surface is flat enough, the measurement error bar is $\sim \pm 6\mu\text{m}$. We made best efforts to choose samples with the (001) surfaces being as flat as possible for Hall measurements. From the device images shown in the inset of Fig. 3e and Fig. S5a and S11a, no clear terraces can be seen on the surfaces of the samples. Of course, we could not exclude small thickness inhomogeneity. The thickness variation less than $6\mu\text{m}$ is not discernable in the microscope. If we take this into account, the measurement error bar of sample thickness should be between $\pm 6\mu\text{m}$ and $\pm 12\mu\text{m}$. As such, for a sample with the thickness of $\sim 100\mu\text{m}$, the sample thickness measurement error should be $\leq 12\%$. As a result, the s value extracted from the QHE could have $\sim \pm 12\%$ error at most due to the measurement error of sample thickness. This explains why the s values of samples E#1 ($s = 2.3$) and E#3 ($s = 2.2$) are slightly larger than the expected value of 2 (see supplementary Table. S1). Note that our estimate of s is based on the assumption that all the conducting Sb layers act as quantum Hall layers. However, some Sb layers may not show QHE due to disorders/defects and/or imperfect contact. If this occurs, it may lead the value of s to be less than 2. We indeed observed this scenario in samples B#1 ($s = 1.5$) and Z#1 ($s = 1.5$). Inhomogeneous transport due to dead layers is often seen in stacked quantum Hall systems such as EuMnBi_2 (Masuda et al., Science Advances 2, e1501117 (2016)). In EuMnBi_2 , the s value extracted from the QHE is $\sim 5-6$, much less than the theoretically predicted value of 8. We have added more discussions on the error of the estimated s to the revised manuscript on page 11 and the thickness measurement information to Methods.

For instance, can equal steps be observed if the Hall conductivity is presented in Fig. 3b? What would be the degeneracy if the conductivity is used for Z#1, which has a large longitudinal resistivity? Why does the longitudinal resistivity become negative at high fields, as seen in Fig. 4a? Will it have any effect on obtaining the correct Hall resistivity?

Response: We present the Hall conductivity σ_{yx} of sample B#1 in Fig. R3b attached below, from which we do not see equal steps, but only one plateau near $B_F/B = 1/2$. However, as shown in Fig. 3b in the manuscript, the inverse Hall resistivity $1/\rho_{xy}$ of this sample, when scaled by the step size of the successive $1/\rho_{xy}$ plateaus (i.e. $1/\rho_{xy}^0$, see the inset to Fig. 3d), i.e. ρ_{xy}^0/ρ_{xy} exhibits equal steps and is quantized to half-integer numbers for $B_F/B = 1/2, 3/2, 5/2$ and $7/2$ (Fig. 3b), indicating QHE. The reason why its σ_{yx} does not display equal steps is that this sample shows an imperfect QHE, as manifested by the non-zero ρ_{xx} at the quantum Hall states. Such an imperfect QHE can be mostly attributed to inhomogeneous transport, as reflected by its smaller degeneracy $s (=1.5)$ estimated from $1/\rho_{xy}^0$.

Nevertheless, we find the sample with higher mobility (E#1) exhibits equal steps in σ_{yx} , as shown in Fig. R3a. This is because that the ρ_{xx} of this sample at the quantum Hall state is very small. For instance, ρ_{xx} is ~ 0.025 m Ω .cm for the ρ_{xy} plateau near 20T (Fig. 4a), which corresponds to the quantum Hall state with the filling factor of $\gamma=3/2$ (Fig. 4c). Such a ρ_{xx} value is one order of magnitude smaller than that of the corresponding $\gamma=3/2$ quantum Hall state of sample B#1 which occurs near 12.5T (see Fig. 3a and 3b in the manuscript). At the quantum state within the quantum limit ($\gamma=1/2$), the ρ_{xx} of sample E#1 becomes much smaller, dropping to zero at about 47.5 T, but turning to slightly negative (~ -0.015 m Ω .cm) above 47.5T (Fig. 4a), as pointed out by the referee.

Fig. R3. Hall conductivity σ_{yx} and longitudinal conductivity σ_{xx} for samples E#1 (a), B#1 (b) and Z#1 (c). Both σ_{yx} and σ_{xx} are obtained from tensor conversions from ρ_{xx} and ρ_{xy} .

Such a negative value is generated by the data symmetrizing, as explained below. The E#1 Hall-bar sample was also prepared through focused ion beam (FIB) cutting. Fig. R4a shows the optical image of this sample. During cooling-down for pulse field measurements on this sample, leads #2 and 4 broke, so we had to use leads #3 and #6 to measure both ρ_{xy} and ρ_{xx} . Fig. R4b presents the raw data of the voltage measured between leads #3 and #6, $V_{3,6}$. Although leads #3 and #6 are significantly misaligned, $V_{3,6}$ show a remarkable asymmetric feature between positive and negative magnetic fields, indicating $V_{3,6}$ is dominated by the Hall voltage V_{yx} and the longitudinal voltage V_{xx} is small. Fig. R4c shows V_{xx} and V_{yx} data obtained through symmetrizing

and anti-symmetrizing of the $V_{3,6}$ data acquired under positive and negative magnetic fields. The V_{yx} plateaus are found to be accompanied by the V_{xx} minima, a typical signature of QHE. The small negative V_{xx} above 47.5T can be attributed to the fact that symmetrizing $V_{3,6}$ between positive and negative fields cannot completely remove the Hall voltage component, which is often seen in Hall effect measurements where the longitudinal and Hall resistivities are mixed. The anti-symmetrizing process of $V_{3,6}$ may also not completely remove V_{xx} from V_{yx} , but the perfect ρ_{xy} plateau near 50T (Fig. 4a) indicates the ρ_{xx} at this quantum Hall state is extremely small. The longitudinal conductivity σ_{xx} for this quantum Hall state is indeed close to zero, as shown in Fig. R3a.

The observations of σ_{yx} equal steps in sample E#1 (Fig. R3a), together with its very small ρ_{xx} and nearly zero σ_{xx} at the quantum Hall state within the quantum limit, suggests its stacked QHE is nearly perfect; that is, almost every 2D Sb conducting layer acts as a quantum Hall layer. Further, we have estimated the degeneracy s from the σ_{yx} plateau at the quantum limit with $B_F/B = 1/2$ using σ_{yx} per Sb layer = $(1/2)s(e^2/h)$; σ_{yx} per Sb layer is derived by dividing the total Hall conductivity σ_{yx} by the sample thickness. The s value obtained through this approach is 2.2, consistent with that estimated from $1/\rho_{xy}^0$ ($s=2.3$); its deviation from the expected value of 2 is apparently due to the errors in the measurements of sample dimensions as explained above. If we take $s=2$ and normalize σ_{yx} per Sb layer by $2e^2/h$, the quantized Hall conductivity plateaus can be seen clearly for $\gamma = B_F/B = 1/2, 3/2, 5/2$ and $7/2$, as shown in Fig. R4d. To clearly demonstrate the nearly perfect QHE of sample E#1, we have added Fig. R4d to Fig. 4d in the main text.

Fig. R4. (a) Optical image of the Hall-bar sample E#1 prepared using FIB. (b) Field dependence of voltage measured between leads #3 and #6, $V_{3,6}$ (see panel a). (c) Hall voltage V_{yx} and longitudinal voltage V_{xx} obtained through anti-symmetrizing and symmetrizing $V_{3,6}$ respectively.

(d) Hall conductivity σ_{yx} per Sb layer, scaled by $\sigma_0=2e^2/h$, as a function of B_F/B for sample E#1(B_F , the SdH oscillation frequency).

As compared to the QHE of sample E#1, the QHE observed in the 10% Zn-doped sample (Z#1) is also imperfect, which is reflected in its large ρ_{xx} values at the quantum Hall states: the minimal ρ_{xx} corresponding to the $\gamma=3/2$ ρ_{xy} plateau near 7.5T is ~ 13.5 m Ω .cm (see Fig. S4a). Such a large ρ_{xx} value leads to absence of plateau in its σ_{yx} (Fig. R3c) despite the presence of ρ_{xy} Hall plateau (Fig. S4b). This result is consistent with the small s value ($s=1.5$) extracted from $1/\rho_{xy}^0$ (Table S1) for this sample, indicating Zn-doping results in the presence of many dead quantum Hall layers. Owing to the absence of σ_{yx} plateau, we could not estimate its s from σ_{yx} . For sample B#1, since we observed one σ_{yx} plateau corresponding to the $\gamma=1/2$ quantum Hall state (Fig. R3b), its s value estimated from the σ_{yx} plateau is 1.6, consistent with that estimated from $1/\rho_{xy}^0$ ($s=1.5$). In the revised manuscript, we have added discussions on the difference of the QHE among samples E#1, B#1 and Z#1 on page 9-10 and added Fig.R3a-3c to the SM (Fig. S3).

I would like to point out that the degeneracy can be obtained by comparing the Hall resistivity, which gives the carrier density, with B_F , which tells the size of the Fermi surface. This is not entirely an independent method, still it is good to show, as it does not have the dilemma of which to choose between resistivity or conductivity.

Response: We thank the referee for this suggestion. We have estimated the carrier density from the quantum oscillation frequency F and compared it with transport carrier density extracted from the Hall coefficient. According to Luttinger's theorem, the carrier density of a 2D system with a degeneracy of 2 can be expressed as $n_{2D} = 2eF/h$ where e is the elemental charge and h is the Planck's constant. Since one-unit cell in BaMnSb₂ contains two conducting Sb layers (Fig. 1a), the 3D carrier density can be expressed as $n_{SdH} = n_{2D}/(c/2)$, where c is the lattice parameter. As shown in Table S1, the estimated n_{SdH} is close to the carrier density extracted from Hall coefficients (n_{Hall}) for samples E#1, E#3 and B#1 by assuming the degeneracy of 2. For instance, n_{Hall} for sample E#1 is $1.4 \times 10^{19} \text{cm}^{-3}$, and its n_{SdH} is $1.24 \times 10^{19} \text{cm}^{-3}$. Such a consistency between n_{SdH} and n_{Hall} gives additional support for the degeneracy of 2. For sample Z#1, we note the difference between n_{SdH} and n_{Hall} is relatively large ($n_{SdH} = 0.36 \times 10^{19} \text{cm}^{-3}$ vs. $n_{Hall} = 0.093 \times 10^{19} \text{cm}^{-3}$), which can be attributed to inhomogeneous transport caused by Zn doping; that is, those Sb layers exhibiting SdH oscillations and QHE have different carrier density from those layers without showing the SdH oscillations and QHE. n_{SdH} represents only the carrier density of those layers showing SdH oscillations, while n_{Hall} is the average carrier density of all layers. We have added these discussions to the revised manuscript on pages 11-12.

As for the plateau in σ_{zz} (Fig. 4b), is it possible that this is due to joule heating when the resistance is strongly enhanced?

Response: The plateau in the z-axis resistance R_{zz} (or conductivity σ_{zz}) near 50T of sample E#2 (Fig. 4b) is a robust feature of the quantum Hall state within the quantum limit, that does not come from joule heating as explained below. We can evaluate the heating effect by comparing the upward and downward field sweep measurements. Fig. R5b attached below shows how the pulse field H varies with time t in upward and downward field sweeps as well as dH/dt (red curve). We present the comparison of R_{zz} measured in the up- and down-field sweeps in Fig.

R5a, from which we can see the R_{zz} peak in the 20-25T range exhibits a striking difference between up- and down-field sweeps. In the main text, we have shown the $\gamma = 3/2$ ρ_{xy} plateau occurs within the 20-25T range (Fig. 4a and 4b). The difference of the R_{zz} peak height between the up- and down-field sweeps within this quantum Hall state should arise from heating effect. In the up-field sweep, the field increase from 20T to 25T takes an extremely short period of time t_1 , such that the heat generated by the field sweep and measurements cannot be dissipated effectively. In contrast, the field decrease from 25T to 20T in the down-field sweep takes much longer time ($t_4 \approx 5t_1$, see Fig. R5b) so that the heating effect can be suppressed, which explains the enhanced R_{zz} peak probed in down-field sweep.

However, the heating effect for the quantum Hall state within the quantum limit ($B > 50T$) becomes much weaker, because the R_{zz} probed above 50T shows much smaller difference between up- and down-field sweeps (Fig. R5a). The time for the field increasing from 50T to 60T (t_2) and the time for the field decreasing from 60T to 50T (t_3) are much longer than t_1 and t_4 (see Fig. R5b), thus the heat generated by magnetic field sweeps and measurements within these time periods is expected to be small.

Furthermore, we did similar up- and down-sweep measurements at 4.1 K and find the hysteresis of R_{zz} due to the heating effect is significantly suppressed for the quantum Hall state at 20-25T, and extremely small for the quantum Hall state near 50T. More importantly, the R_{zz} values at 4.1 K and 0.7K are nearly identical for fields close to 60T. If the R_{zz} plateau was due to heating effect, we would expect the R_{zz} value near 60T to decrease as the temperature increases to 4.1K, inconsistent with the observation of nearly identical R_{zz} at 0.7 K and 4.1K for fields approaching 60T. The evolution trend of the R_{zz} plateau from 0.7K to 4.1K also implies that the R_{zz} plateau should become more flattened as the temperature is further decreased below 0.7K. All these facts indicate that the R_{zz} plateau at the quantum Hall state within the quantum limit should be intrinsic and implies the presence of 2D chiral surface state as discussed in the manuscript. We have added these discussions as well as Fig. R5 into the SM (section I.7 in SM).

Fig. R5: (a) The z -axis resistance R_{zz} measured in the up- (red/purple) and down-field (blue) sweeps for sample E#2 at 0.7 K and 4.1 K. (b) The variation of pulse field with time and the derivative (red) of the field relative to time. t_1 (t_4) represents the time period for the field increasing (decreasing) from 20T (25T) to 25 T (20T). t_2 and t_3 represent the time periods of the field sweep from 50T to 60T and then from 60T to 50T respectively.

REVIEWER COMMENTS

Reviewer #2 (Remarks to the Author):

The authors have carefully addressed the comments made by referees. The main concern I had is the reliability of the estimation of the Landau level degeneracy. In the reply and revision, more data are provided, showing additional evidence and the consistency among samples. They have carried out corresponding discussions, which I find are convincing. I support its publication in Nature Communications.

Reviewer #3 (Remarks to the Author):

This work reports the observation of a 3D quantum Hall effect in BaMnSb₂ resulting from a spin-valley locked band dispersion. The work additionally argues that chiral side-surface states are observed in the quantum limit, as has been reported in the past in III-V superlattices. Novelty issues were raised by previous referees given that a prior work submitted in March 2019 demonstrated similar results [Phys. Rev. B 101, 081104(R), (2020)]. However, by the editor's request, I will overlook this concern. In their response to other referees, the authors do highlight additional characterization beyond what was shown in [Phys. Rev. B 101, 081104(R), (2020)], which further justifies the novelty of the content. The manuscript should be published but I have some questions that I think the authors need to address first:

1. By analogy to III-V superlattices, the authors conclude that a plateau in R_{zz} is indicative of chiral side surface states. The authors have discussed how to rule out other contributions such as trivial surface states and the presence of an additional Fermi surface. Related to that last point, can the authors discuss why they can specifically rule out charge transfer between the X(Y)-valleys and Gamma-valley, in light of what has been observed in graphene or bismuth by the Behnia group?
2. In the Landau level model, the expression for the $N=0$ levels contains a term proportional to $v_1 v_2 / l_B^2$. Is that term due to quadratic terms in Hamiltonian?

We thank both referees for reviewing our manuscript. We appreciate their comments and suggestions, which have been very helpful in improving our manuscript. In the following, we provide a point-to-point response (shown in blue) to the comments raised by Referee 3 (shown in black).

Response to Reviewer #3

This work reports the observation of a 3D quantum Hall effect in BaMnSb₂ resulting from a spin-valley locked band dispersion. The work additionally argues that chiral side-surface states are observed in the quantum limit, as has been reported in the past in III-V superlattices. Novelty issues were raised by previous referees given that a prior work submitted in March 2019 demonstrated similar results [Phys. Rev. B 101, 081104(R), (2020)]. However, by the editor's request, I will overlook this concern.

In their response to other referees, the authors do highlight additional characterization beyond what was shown in [Phys. Rev. B 101, 081104(R), (2020)], which further justifies the novelty of the content. The manuscript should be published.

Response: We thank the referee for taking time to review our manuscript and appreciate the referee's judgment: "The manuscript should be published".

but I have some questions that I think the authors need to address first:

1. *By analogy to III-V superlattices, the authors conclude that a plateau in R_{zz} is indicative of chiral side surface states. The authors have discussed how to rule out other contributions such as trivial surface states and the presence of an additional Fermi surface. Related to that last point, can the authors discuss why they can specifically rule out charge transfer between the X(Y)-valleys and Gamma-valley, in light of what has been observed in graphene or bismuth by the Behnia group?*

Response: We thank the referee for raising this question and drawing comparison to the Behnia group's studies of graphene and bismuth in the magnetic quantum limit where charge transfer between bands has a profound influence on the quantum oscillation spectrum. This work highlights that charge transfer between valleys can lead to non-trivial Landau level evolution (Zhu et al., Nature Communications 8,15297, 2017), thus resulting in a striking change in magnetoresistance.

For the material studied in our work, BaMnSb₂, although the lower photon energy ARPES shown in the SI indicates multiple valleys (an additional trivial pocket at Γ), we can exclude the possibility of the charge transfer between the X(Y) and Γ valleys leading to non-trivial Landau level evolution, as discussed below. First, the background of the field dependence of Hall resistivity ρ_{xy} nearly shows temperature independent linear behavior if we do not consider the ρ_{xy} plateau above 20T (see Fig. 3f in the manuscript). This low field linearity, combined with the higher field plateaus, originating from the X-point pockets, indicates that the in-plane transport is dominated by a single band despite the existence of the band near the Γ point. Second, the SdH oscillations exhibit a single frequency. As shown in Table S1 in the supplemental materials, the carrier density (n_{SdH}) estimated from the quantum oscillation frequency is close to the carrier density extracted from Hall coefficients (n_{Hall}) for most of the samples used in this study (i.e. samples E#1, E#3 and B#1) [Note that there is only one exception for sample Z#1 and this sample exhibits a relatively large difference between n_{SdH} and n_{Hall} ,

which can be attributed to inhomogeneous transport caused by Zn doping, as discussed in the manuscript on page 12]. The above facts, together with the spin-valley degeneracy analyses based on the observed quantum Hall effect (Fig. 3 and 4) and the Landau level spectrum calculations (Fig. 4f), indicate that the high mobility Dirac fermions hosted by X(Y) valleys are responsible for the observed SdH oscillations and quantum Hall effect, and that the Γ valley has negligible contributions to transport and does not show quantized Landau levels even under high magnetic fields due to a significantly lower mobility. This is not only consistent with the calculated band structure (Fig. 2a) which shows the X(Y) valley is characterized by linear Dirac bands, whereas the Γ valley shows nearly flat dispersion at the valence band top. The low carrier mobility of the Γ valley also explains why this Fermi pocket is not probed in the SdH oscillations.

Given that the carriers of the X(Y) valleys have much higher mobility than those of the Γ valley (which does not exhibit Landau quantization in this field range), the only effect of any Landau quantization driven by charge transfer would be the extent to which the Landau quantization of the X(Y) valleys follows a canonical vs grand canonical description, i.e. the evolution of Landau levels with a fixed particle number and oscillating chemical potential in the former, or fixed chemical potential and oscillating particle number in the latter. By contrast, the work of Behnia et al exhibits non-trivial Landau Level evolution because both sets of pockets involved (for both Bismuth and Graphite) are strongly Landau quantized and approaching their magnetic quantum limits. This is apparently not what we observed in the field range where the R_{zz} plateau is present (Fig. 4b). As shown in Fig. 4b, when R_{zz} reaches a plateau, ρ_{xx} drops to a minimum close to zero. More importantly, as seen in Fig. 4d, in the field range where the R_{zz} plateau is observed, the Hall conductivity σ_{yx} shows the $\gamma = 1/2$ plateau and the corresponding σ_{xx} is close to zero. This clearly demonstrates the system is in a nearly ideal quantum Hall state within the quantum limit. These results indicate the R_{zz} plateau is the consequence of the quantum Hall state and suggestive of the chiral surface states as discussed in the manuscript.

We have added some of the above discussions to the revised manuscript on page 15-16 and cited the Behnia group's work on the charge transfer of bismuth.

2. *In the Landau level model, the expression for the $N=0$ levels contains a term proportional to $v_1 v_2 / l_B^2$. Is that term due to q quadratic terms in Hamiltonian?*

Response: Yes, the term proportional to $b_0 |v_1 v_2| / l_B^2$ in the $N=0$ Landau levels originates from the q -quadratic term (See the b_0 term in Eq. (29), (31), (35) and (36) in Supplementary Materials).